# Arctic Ocean virus communities and their seasonality, bipolarity, and prokaryotic associations

Alyzza M. Calayag[1,2], Taylor Priest [3], Ellen Oldenburg [4,5,6], Jan Muschiol [1], Ovidiu Popa[5], Matthias Wietz [4,7,8] & David M. Needham [1,2] ✉

Viruses of microbes play important roles in ocean environments as agents of mortality and genetic transfer, influencing ecology, evolution and biogeochemistry. However, we know little about the diversity, seasonality, and host interactions of viruses in polar waters. Here, we study dsDNA viruses in the Arctic Fram Strait across four years via 47 long-read metagenomes of the cellular size-fraction. Among 5662 vOTUs, 98% and 2% are *Caudoviricetes* and *Megaviricetes*, respectively. Viral coverage is, on average, 5-fold higher than cellular coverage, and 8-fold higher in summer. Viral community composition shows annual peaks in similarity and strongly correlates with prokaryotic community composition. Using network analysis, we identify putative virus-host interactions and six ecological modules associated with distinct environmental conditions. The network reveals putative novel cyanophages with time-lagged correlations to their hosts (in late summer) as well as diverse viruses correlated with *Flavobacteriaceae*, *Pelagibacteraceae*, and *Nitrosopumilaceae*. Via global metagenomes, we find that 42% of Fram Strait vOTUs peak in abundance in high latitude regions of both hemispheres, and encode proteins with biochemical signatures of cold adaptation. Our study reveals a rich diversity of polar viruses with pronounced seasonality, providing a foundation for understanding viral regulation and ecosystem impacts in changing polar oceans.

Polar regions are subject to the strongest seasonal cycles on Earth, and experience intense pressure from climate change[1,2]. The functioning of polar ecosystems is critical to biogeochemical cycles[3–5], and is under marked ecological and evolutionary constraints[6]. Such ecosystem processes are strongly driven by microorganisms and their interactions, including viral dynamics. In addition to cell death, viruses also drive evolution via gene exchange, frequency-dependent selection[7,8], and transmission and expression of auxiliary metabolic genes[9–11].

Therefore, characterizing the diversity and dynamics of viruses is paramount for understanding the function and stability of polar ocean ecosystems.

Relative to more accessible temperate, subtropical, and tropical oceans, few studies have examined virus diversity and ecology in polar waters[12,13]. It is known, however, that polar viromes are distinct from their warm-water counterparts[13–22] and are typically dominated by bacteriophages with a high level of diversity[14]. In addition to spatial

[1]GEOMAR Helmholtz Centre for Ocean Research Kiel, Kiel, Germany. [2]Faculty of Mathematics and Natural Sciences, Kiel University, Kiel, Germany. [3]Institute of Microbiology, ETH Zurich, Zurich, Switzerland. [4]Alfred Wegener Institute Helmholtz Centre for Polar and Marine Research, Bremerhaven, Germany. [5]Institute of Quantitative and Theoretical Biology, Heinrich Heine University, Düsseldorf, Germany. [6]Cluster of Excellence on Plant Sciences, Heinrich Heine University, Düsseldorf, Germany. [7]Max Planck Institute for Marine Microbiology, Bremen, Germany. [8]Institute for Chemistry and Biology of the Marine Environment, University of Oldenburg, Oldenburg, Germany. ✉e-mail: dneedham@geomar.de

structuring, polar viral communities also shift over time, for example in the Antarctic, following bloom dynamics and assembling into distinct communities across seasons[23]. One year-round study showed strong seasonal variation among virus communities, with sharp decreases in virus-to-prokaryote ratios at the onset of the spring bloom and highest ratios in winter[22]. Others have reported increases during spring-summer, and highest abundances of viruses in winter[24,25]. Some evidence suggests that different lifestyles of viruses (i.e., lytic vs. lysogenic) may exhibit distinct dynamics across seasons, with lytic infection more prevalent in the spring bloom[26].

However, due to the challenges of continuous sampling in the polar regions across multiple years, the degree of seasonality among polar viruses—here, meaning annually repeating patterns of populations and communities across the same seasons of different years[27]— and the potential ecological implications remain to be discovered. Such seasonality has been previously observed in marine temperate environments[28–30]. In the Arctic, time-series have been critical to advancing an understanding of the biological carbon pump[31], elucidating benthopelagic coupling and biotic interactions[32], both in the water column[33] and on sinking particles[34]. Furthermore, continuous sampling, over the long-term, can discern the impact of 'Arctic Atlantification'[35–37]: the northward expansion of subarctic habitats through the Fram Strait—the major connection between Atlantic and Arctic Oceans[38]. Here, polar water outflowing the central Arctic Ocean via the East Greenland Current (EGC) meets the West Spitsbergen Current (WSC), transporting warmer Atlantic water into the Arctic Ocean[39–41]. In the WSC, prokaryotic communities exhibit pronounced seasonality, underpinned by changes in photosynthetically active radiation (PAR) and mixed layer depth[42].

Given this, and the tight coupling of viruses and their hosts[9–11], we aim to examine the degree that viral populations are seasonally structured in the Arctic. The strong seasonal gradients in the Arctic provide an opportunity to observe host-virus dynamics and how they relate to prevailing conceptual models such as the Piggyback-the-winner[43–45], Constant-Diversity[46,47], and Red-Queen[29,48] hypotheses. We examine the diversity and seasonality of dsDNA virus communities in the Arctic Fram Strait over four complete annual cycles at roughly monthly resolution. This time-series of virus communities in the Arctic demonstrates considerable seasonality in viral diversity, their association with environmental conditions and potential microbial hosts, and their distribution across the global ocean.

## Results
We examined 47 long-read metagenomes from samples collected at near-monthly resolution over a four-year period (Sep 2016–Jul 2020)

in the WSC (Fig. 1a, and Supplementary Data 1), from an average depth of 29 m. As samples originate from the cellular size-fraction (> 0.2 μm), our study characterizes the diversity of actively infecting (i.e., intracellular) viruses, free-living viruses with a size of > 0.2 μm, viruses attached to particles, and/or integrated viruses.

The total number of reads per sample were $196,489 \pm 18,358$, with an average length of $5435 \pm 405$ bp. Viral sequences were predicted based on both mapping to assembled contigs and raw reads (see Virus Prediction, Methods). The overall number of predicted viral sequences was concordant between the two approaches (Supplementary Data 1). On average, each sample harbored $9815 \pm 965$ viral reads based on raw read counts ($5.3 \pm 0.5\%$), and $8825 \pm 1020$ reads mapping to vOTUs (viral Operational Taxonomic Units) ($6.8\% \pm 1.8\%$). Notably, despite not pre-filtering to remove eukaryotes, the large majority of cellular reads in the dataset were prokaryotic ($90.2\% \pm 1.4\%$), with eukaryotic reads making up a smaller proportion ($7.2\% \pm 1.0\%$). Hereafter, we focus on the contig-based predictions because of their improved genomic context.

### Diversity of Fram Strait viruses over time
We first investigated how the community of Fram Strait viruses is structured over time. Through contig-based predictions, we identified 5662 vOTUs (95% nucleotide identity clusters, see Methods, Supplementary Data 2) which were derived from 7775 viral contigs that were greater than 10 Kb in length. These vOTUs predominantly represented the classes *Caudoviricetes* and *Megaviricetes*, with *Caudoviricetes* being the largest group, comprising 5531 vOTUs.

To assess the temporal dynamics of vOTUs, we normalized their abundance based on the estimated number of cells sequenced in each metagenome—a metric we term coverage-based Virus to Cell Ratio (cVCR). The cVCR approach not only accounts for differences in sequencing depth but also captures shifts in virus to host ratios across samples. Using this metric, we observed clear seasonal structuring in the cVCR of viral communities and taxa. Community cVCR values reached annual maxima of 20–35 and were four-fold higher during July–September (average cVCR 8.6) than during October–June (Fig. 1b). These patterns were consistent also when we sub-sampled the total reads to 100,000 reads per sample, as a test of the robustness of this metric to sequencing depth, as well as when we calculated coverage per Gigabase (Supplementary Fig. 1), thus we focus the remaining analyses on cVCR from the full dataset. This seasonal variation in cVCR was generally consistent across *Caudoviricetes*. Notably, the most abundant vOTU, a member of the *Caudoviricetes*, was also the most persistent, occurring in 45 of the 47 sampled time points. Similarly, the most persistent vOTUs overall, present in half of the samples,

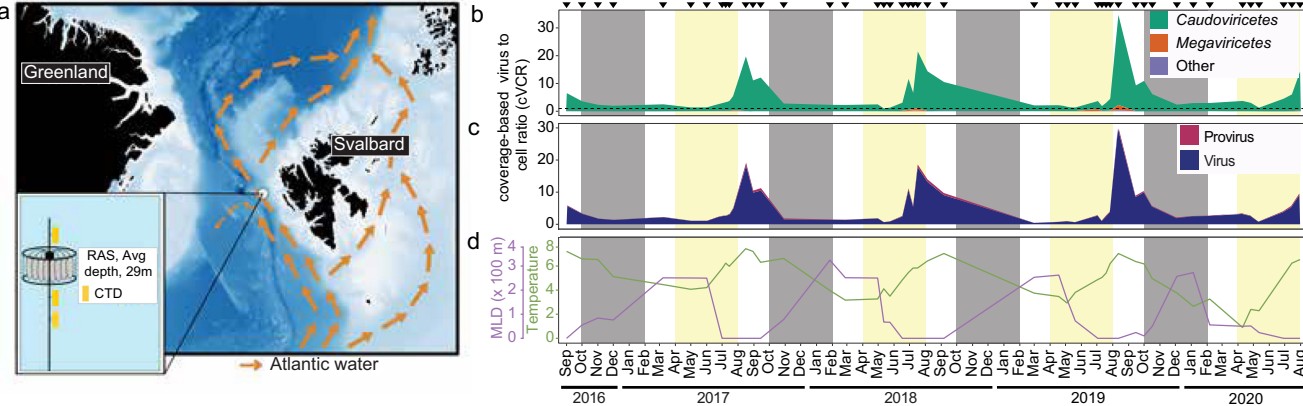

**Fig. 1 | Overview of study site, environmental conditions and viral distributions. a** The mooring site in the West Spitsbergen Current of Fram Strait. **b** cVCR of viral communities across the two major classes detected and (**c**) lifestyles during the light and dark cycles from September 2016 to July 2020. **d** Dynamics of mixed layer depth and temperature which are major community structuring factors. Triangles (top) indicate metagenome sampling points.

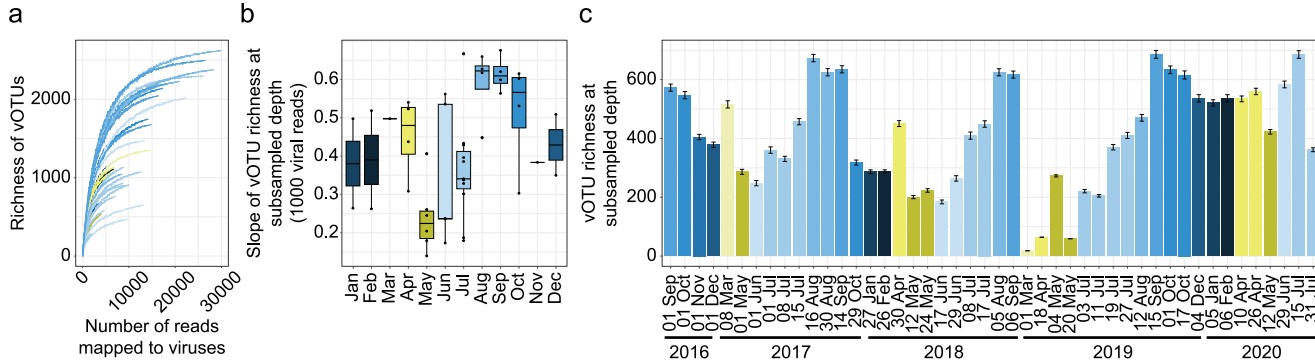

**Fig. 2 | Intra-and inter-annual viral richness variability. a** To assess differences in sequencing depth, we iteratively subsampled viral read counts from 50 up to 30,000 at 50 count intervals and at each interval determined the mean richness from 100 iterations. The mean richness across subsampled intervals was visualized in a rarefaction-style curve. **b** Boxplots illustrating the slope of the vOTU richness rarefaction curves up until the chosen subsampling depth (1000 viral reads) where each box encompasses the 25th, median, and 75th percentile, and the whiskers capture the minimum and maximum values of richness estimated from samples collected within the same sampling month. The slopes represent the trajectory of vOTU discovery, and thus indicate how richness may appear if a higher sequencing depth was achieved. The boxplots illustrate the 25%, median and 75% percentile of richness estimations for samples within each sampling month. **c** vOTU richness at subsampled depth across sampling months. Illustrated values represent the mean (bars) and standard error (error bars) of richness determined from 100 iterations of subsampling (1000 viral read counts per sample).

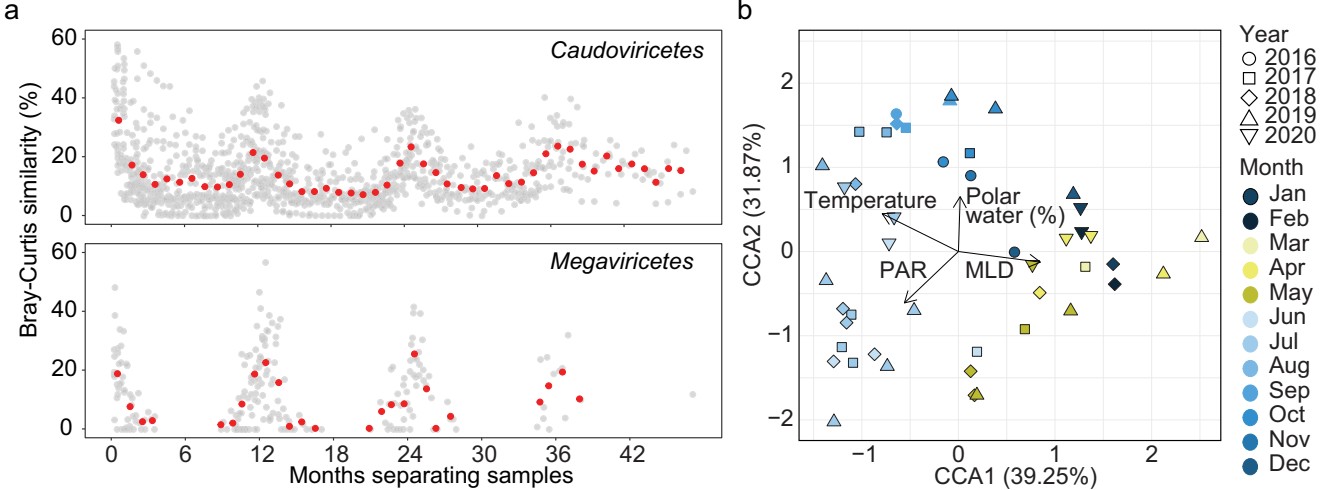

**Fig. 3 | Seasonality of viral communities and their association with environmental conditions. a** Bray-Curtis similarity of *Caudoviricetes* and *Megaviricetes* over time. Each point represents the time between two individual sampling points (*x*-axis) and their similarity (*y*-axis) for all pairs of samples. The red points indicate the average similarity for 30.5-day (-monthly) intervals. **b** CCA analysis of viral community composition colored by month, with vectors representing environmental conditions.

were predominantly members of the *Caudoviricetes*. In contrast, *Megaviricetes* were found less frequently, but were prevalent in late summer (July–September) (Fig. 1b, and Supplementary Fig. 1a), when eukaryotic phytoplankton, their presumed hosts, are most abundant[49]. Overall, lytic lifestyle was predicted to be predominant, with an average of 94.0 ± 0.4% of viruses, a pattern that was invariable across seasons (Fig. 1c). The Southern Ocean featured higher rates of lysogeny during periods of low production[26]. As for the latter point, it is important to point out that some lysogens may not be recognized due to missing integrases or other factors, and, vOTUs may not represent the full prophage, which could then be misinterpreted as lytic rather than integrated. Additionally, lytic viruses may outnumber lysogens in terms of copies within the cellular metagenome, making them easier to detect and assemble, which could bias the recovery or obscure lysogens.

We next investigated how the overall diversity of viruses changes across seasons. As viral diversity was not saturated at any sequencing depth (Fig. 2a), we compared the diversity of vOTUs across months after subsampling to a normalized depth. The diversity of viruses reached a maximum between late summer-autumn (Aug–Oct) (Fig. 2b, Supplementary Fig. 2), with lowest richness values typically in May. These patterns were consistent both by estimates from the slopes of rarefaction curves to the sub-sampled depth (Fig. 2b) as well as the extrapolated richness (Supplementary Fig. 2). Notably, prokaryotic community richness was also lowest in May, though the timing of highest prokaryotic richness was different than for viruses, with the highest prokaryotic richness observed in winter (vs. late summer-autumn for viruses)[42] (Supplementary Data 1). Inter-annual variability in richness was also observed, for example, with May 2019 showing elevated richness relative to other years (Fig. 2c). The dynamics in richness calls for further sequencing efforts, with increased depth of sequencing to further unravel the viral diversity that persists, especially, through late polar night.

In addition to the variations in overall relative abundance (cVCR) and diversity, we also observed strong seasonality in the composition of the major bacteriophage groups (Fig. 3a). The composition of

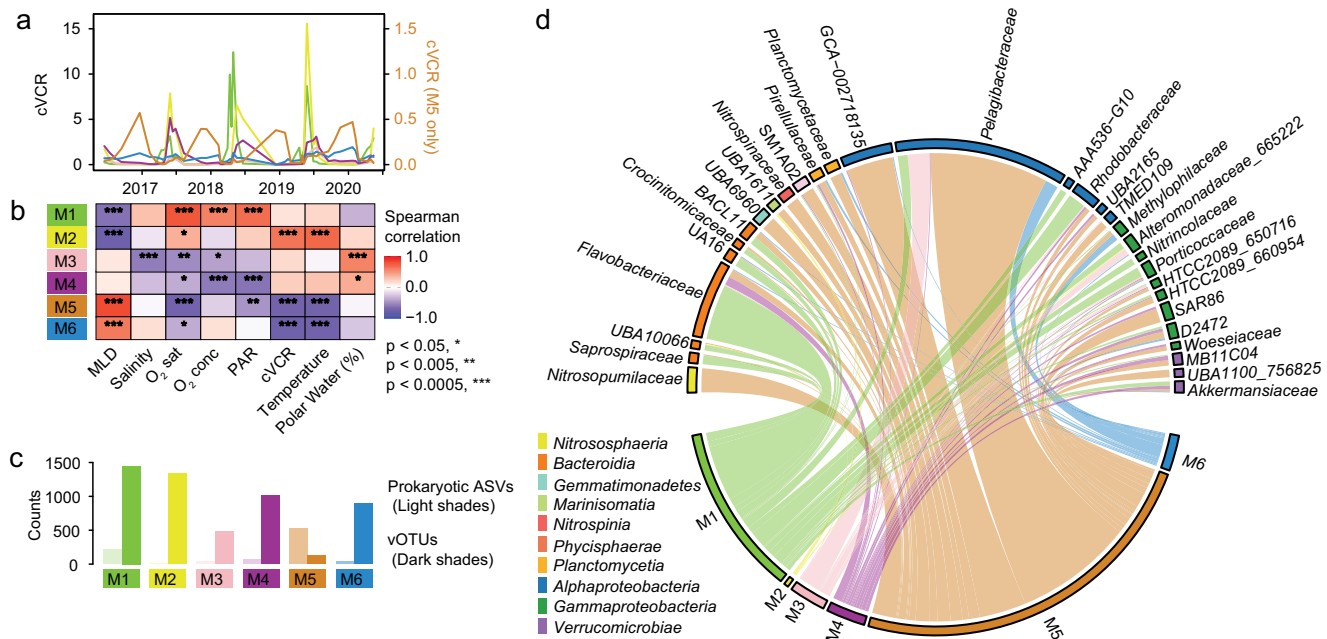

**Fig. 4 | Viral modules and their association with environmental conditions.**
**a** Sum of cVCR of all vOTUs per module over time; note separate *y*-axis for M5.
**b** Two-sided Spearman's rank correlation coefficients between abundances (cVCR) of viral modules and environmental parameters, with significance levels of correlations indicated by asterisks. **c** Count of prokaryotic and viral members of each major module. **d** Chord diagram showing the taxonomy of bacterial ASVs in each module. The taxa along the arc are colored based on their corresponding phylum.

vOTUs within both *Caudoviricetes* and *Megaviricetes* showed a sinusoidal-like pattern over time, with peaks in Bray-Curtis similarity at 12, 24, and 36 month intervals (Fig. 3a). Furthermore, in some cases, similarity between samples from opposing seasons was zero, which is likely a combination of strong seasonality, low relative abundance (and hence detection) of vOTUs in winter, as well as variable sequencing depth. Notably, the seasonality observed in virus community composition was consistent regardless of minimum coverage value utilized for detected viral presence, though similarity was less at higher thresholds (Supplementary Fig. 3).

To further contextualize the dynamics at the vOTU level, we examined the underlying sequence diversity of vOTUs in two ways. First, we explicitly compared all non-clustered viral contigs > 10,000 bp across years. Out of 7360 such contigs, we found 21 clusters of contigs, encompassing 55 contigs total that were 100% matches between at least two years. This suggests that detection of viruses with no genetic changes across years is rare, though there is a limitation of sequencing depth. Second, based on analysis of single nucleotide variations in reads to vOTUs (via InStrain, see Methods), we find that there tends to be microdiversity underlying the vOTU populations, though the amount varies between vOTUs ranging from likely insignificant and potentially related to sequencing error, to high differentiation (e.g., 97% similarity) (Supplementary Fig. 4). These analyses highlight that, in general, vOTUs are made up of populations of highly similar viruses, rather than being clonal. This has also been observed elsewhere, including the Arctic[14].

### Virus-host and environmental relationships

To explore the temporal structuring of viruses and their association with prokaryotic hosts, we employed community- and taxon-level analyses in the context of eight physicochemical parameters and prokaryotic community composition data, comprising 3748 prokaryotic amplicon sequence variants (ASV).

At the whole community level, Mantel tests demonstrated the strongest correlation to prokaryotic community composition (Mantel $\rho = 0.632$, $p = 0.001$, $n = 46$) followed by oxygen (Mantel $\rho = 0.371$,

$p = 0.001$, $n = 46$) and mixed layer depth (MLD) (Mantel $\rho = 0.259$, $p = 0.001$, $n = 46$) (Supplementary Data 3, Supplementary Fig. 5). To further elucidate physicochemical drivers, we used canonical correspondence analysis (CCA) to reveal a seasonal clustering of samples, with temperature (summer), MLD (winter), photosynthetically active radiation (PAR; late spring-early summer), and polar water fraction (late summer) accounting for 15.4% of the total variance (Fig. 3b). These results suggest that viral communities are primarily correlated with prokaryotic community composition which is in turn driven by environmental conditions plus biological interactions—leading to a complex network of interdependencies and physicochemical linkages, similar to dynamics in temperate environments[7,50].

Given the coupling between viral and prokaryotic communities, we next explored potential virus-host associations over time at the individual vOTU and 16S rRNA gene ASV level. We constructed a Convergent Cross Mapping (CCM) network based on co-occurrences, similar to the approach described in Oldenburg et al.[49], which includes information about the direction of associations (i.e., predicting causal relations) between vOTUs and ASVs. The CCM network comprised 5136 vOTUs and 850 prokaryotic ASVs (Supplementary Fig. 6a), with directional associations (correlations > 0.7) where vOTUs dynamics are 'following', or are 'caused' by, prokaryotic ASV dynamics (akin to Lotka-Volterra dynamics[51]) totaling 15,930. Louvain clustering resulted in six distinct modules, comprising vOTU-ASV associations occurring during the same temporal period. We named the six modules based on their seasonal abundance patterns (Fig. 4a); for example, M1 peaks in late spring, followed by M2. The temporal distinction between modules was confirmed by their correlations with specific physicochemical conditions (Fig. 4b). M1 showed the strongest positive correlation with PAR, while M2 had the strongest positive correlation with temperature (Fig. 4b). Interestingly, these two modules also have the most diverse virus communities, with M1 containing 1448 and M2 containing 1334 vOTUs, respectively (Fig. 4c). M3 and M4 peaked in late summer, with M3 having the strongest positive correlation with polar water fraction, and M4 having the strongest negative correlation with PAR. M5 and M6 both negatively correlated with temperature, cVCR, and positively with

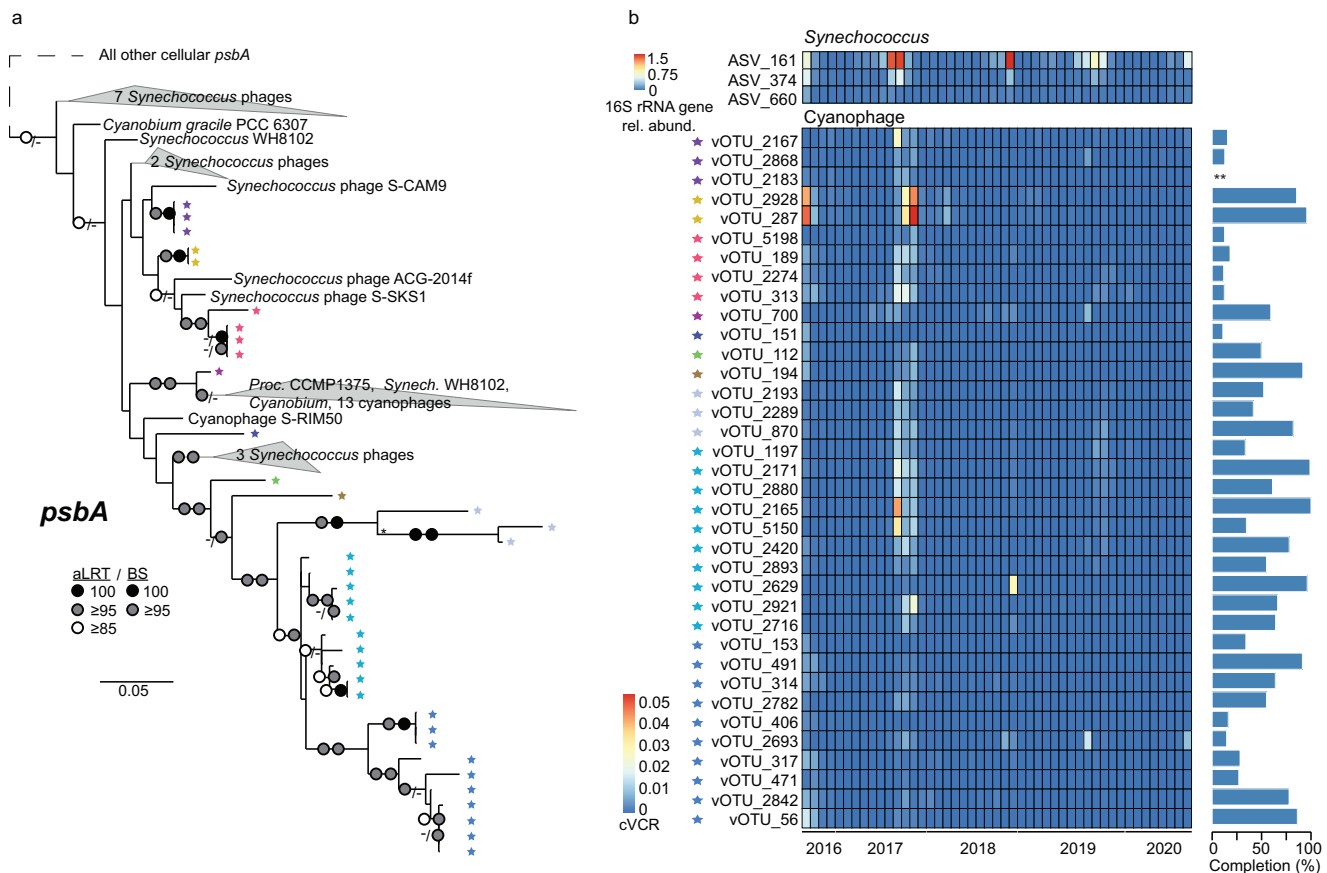

**Fig. 5 | Cyanophages. a** *psbA* gene phylogeny of cyanophage vOTUs and reference genomes. Support values are reported as aLRT (approximate likelihood ratio test) / BS (maximum likelihood bootstrap). For visualization purposes, "*" indicates branches from this node are scaled 33% of their actual length. **b** Dynamics of correlated *Synechococcus* ASVs (top) and cyanophage vOTUs (bottom), along with completion estimates based on CheckV[108]. ** indicates no completion prediction as the vOTU was classified as a provirus.

MLD. However, among these two, only M5 negatively correlated with PAR, being particularly abundant during winter (December–April), while M6 was persistent throughout the year. Generally, the number of vOTUs in a given module outnumbered the number of prokaryotic ASVs, except M5 where ASVs outnumbered vOTUs. Notably, M5 also contained the fewest vOTUs, 132 (Fig. 4c). The overall trend of more vOTUs than ASVs in modules corresponds to the overall larger number of vOTUs vs. prokaryotic ASVs examined.

The modules were consistently dominated by *Caudoviricetes* (Supplementary Fig. 6b), and the proportion of lytic to lysogenic viruses was rather invariable (Supplementary Fig. 6c). *Megaviricetes* were primarily present in the spring and early summer modules M1 and M2 (Supplementary Fig. 6b) where they constituted 6.3% and 1.4% of the vOTUs, respectively. Together, the major differences between modules, especially relating to bacteriophages, are hence at the vOTU level.

In terms of prokaryotic membership, the modules varied considerably. M1 was dominated by taxa typically associated with copiotrophic conditions or phytoplankton blooms, including *Flavobacteriaceae*, *Rhodobacteraceae*, and *Porticoccaceae* (Fig. 4d). In contrast, M5, which contained the largest number of ASVs, comprised diverse prokaryotic taxa, including *Pelagibacteraceae*, *Nitrosopumilaceae*, GCA−002718135 (aka HIMB59), *Nitrospinaceae*, SAR86, *Pirellulaceae*, and *Planctomycetaceae* (Fig. 4d). The smaller modules M2, M3, M4, and M6 comprised a variety of taxa, with the most prevalent taxon within each being UBA1611 (Marinimicrobia), and *Pelagibacteraceae*, respectively (Fig. 4d).

In order to evaluate host association patterns, we utilized both the CCM network correlations and host predictions via iPHoP[52]. In total, 42.5% of vOTUs received a host prediction by iPHoP. Among bacterial families with a high number of predicted vOTUs, *Flavobacteriaceae* and *Pelagibacteraceae* were the most frequently predicted. In M1, *Flavobacteriaceae* and *Akkermansiaceae* (Verrucomicrobia) dominated, while *Pelagibacteraceae* was less common. Conversely, in M5, *Pelagibacteraceae* was the most predominant, with fewer *Flavobacteriaceae* and *Akkermansiaceae*. Meanwhile, vOTUs with *Cyanobiaceae* as their predicted host were more common in M2–M4 (Supplementary Fig. 6d). Among the vOTUs correlated with an ASV in the network, the majority of host-virus pairwise correlations were to diverse prokaryotes present in M5, with *Pelagibacteraceae* (20.3%) and *Nitrosopumilaceae* (6.0%) being the most common (Supplementary Fig. 6e). M1 had the second most bacteria-to-virus connections with 37.5% of these being to *Flavobacteriaceae*. Among the families with most pairwise correlations (i.e., over 250), the overall pattern of host prediction of the viruses and module membership was similar, with vOTUs with predicted hosts of *Flavobacteriaceae* and *Pelagibacteraceae* being the dominant host predictions for M1 and M5, respectively (Supplementary Fig. 6f). However for individual vOTUs and ASVs, only 3.7% of the predictions overlapped at the family level between CCM network correlations and iPHoP (i.e., 122 out of 3,280 correlations with associated host predictions). Among these, again, *Flavobacteriaceae* and *Pelagibacteraceae* made up 46.7% and 6.6%. For *Flavobacteriaceae*, these vOTUs were diverse, coming from 14 different novel orders (via VConTACT3[53], see Methods), while for *Pelagibacteraceae* they all came

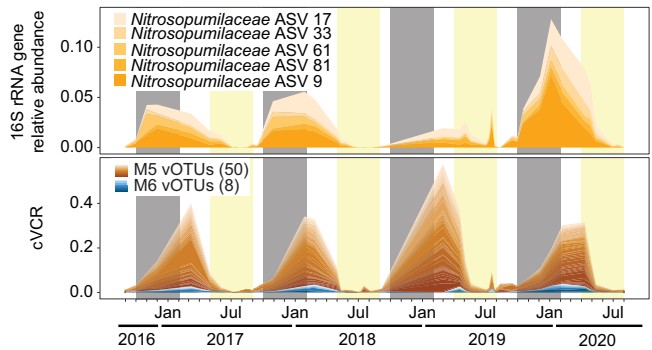

**Fig. 6 | Dynamics of *Nitrosopumilaceae* and associated viruses.** The shown *Nitrosopumilaceae* are the most abundant ASVs, which are all members of M5. Different ASVs and vOTUs are shown by different shades of orange and blue, corresponding to M5 and M6 colors, respectively. The shown vOTUs are those that are correlated with *Nitrosopumilaceae* ($p < 0.05$) in the Convergent Cross Mapping network (see Methods, Supplementary Fig. 5a, 6).

from a single novel order (Supplementary Data 4). However, overall, the low number of matching CCM correlations and host predictions at the vOTU-ASV level demonstrates the challenge to discern host-virus relationships in diverse and complex ecosystems where a variety of technical and biological challenges may complicate such analyses (see Discussion). Nonetheless, the results provide valuable indications for particular lineages at an overall module level.

## Diversity and dynamics of cyanophages and *Nitrosopumilaceae*-associated viruses

From the predicted interactions between vOTUs and ASVs, we further explored the diversity and dynamics of putative cyanophages—as cyanobacteria are of particular interest with respect to changing conditions in Arctic ecosystems[5,54,55]. We identified putative cyanophage vOTUs based on phylogenetic analysis of *psbA* genes, classification based on VPF-Class[56] and host prediction via iPHoP[52]. The phylogenetic reconstruction of *psbA* revealed that putative cyanophages are distinct from cultivated relatives from more temperate locations (Fig. 5a). The putative cyanophage vOTUs represented some of the largest viral contigs recovered, with half of the assembled cyanophages being medium- to high-quality, and a quarter >80% complete (Fig. 5b, and Supplementary Data 2). The cyanophage vOTU abundance peaked during August–September 2017 (Fig. 5b), complementing previous observations of *Synechococcus* in the WSC[42,54]. Although the seasonal dynamics were consistent across the cyanophage vOTUs, their maximal cVCR varied from 0.003 to 0.06 (Fig. 5b). In addition, all of the cyanophage vOTUs exhibited lower abundances in 2018–2020 compared to 2017, indicating interannual variation of these viruses and their presumed hosts (Fig. 5b). Overall, the abundances of the cyanophage vOTUs with *Synechococcus* ASVs were correlated when considering no time-lag (Spearman's $\rho = 0.48$, $p = 0.0008$, $df = 44$), but a stronger correlation (Spearman's $\rho = 0.51$, $p = 0.0008$, $df = 44$) occurred with a vOTU time-lag of one time-point (average interval of 32 days). Thus, putative cyanophages increased in abundance ~1 month after *Synechococcus* peaks. For individual ASVs and vOTUs, the strongest correlations were without time delay ($n = 53$), followed by one ($n = 44$) and two ($n = 32$) time-points, indicating some temporal variability between individual cyanophage vOTUs and their putative hosts compared to the groups at-large (Supplementary Data 5). Our results reveal that both cyanobacteria and their viruses are present in the Arctic, highlighting a need to further understand their interactions for the future Arctic Ocean, as they are expected to increase due to Atlantification.

Nevertheless, given the scarcity of data on viruses during the polar night, we also examined the microbial dynamics of viruses related to

prokaryotes dominant in winter. To do so, we focused on viruses associated with *Nitrosopumilaceae* due to their importance for wintertime nitrogen and carbon cycling[57,58], their prevalence in the dataset with the top five *Nitrosopumilaceae* ASVs peaking from October–May (average 4.2%) vs. lower abundances from June–September (average 0.5%), and being among the taxa with the most correlations to viruses (Supplementary Fig. 7). We identified associations between 58 vOTUs and five *Nitrosopumilaceae* ASVs (Fig. 6), which were primarily associated with the winter module M5. In total, 53 of the 58 vOTUs were classified as *Caudoviricetes* (the other five were unclassified), and came from 14 different novel orders, based on vConTACT3 (Supplementary Data 6). Notably, among vOTUs with correlations to hosts in the correlation network, only one had a predicted host (based on iPHoP) of *Nitrosopumilaceae*. This vOTU was correlated with an unclassified Gammaproteobacterium (no family-level prediction) within the correlation network, highlighting again the difficulty comparing directly these two complex and independent approaches. The persistence of these vOTUs despite low virus cVCR demonstrates the ability to track pronounced seasonality even amongst low abundance viruses. However, more focused investigations on the temporal linkages of these vOTUs and their potential hosts are needed to better understand the environmental impacts of such associations.

## Bipolarity of Fram Strait viruses

Considering the long-standing discussion on latitudinal microbial diversity gradients[59–62] and the endemicity of Arctic and Antarctic microbiomes[63,64], we assessed the distribution of Fram Strait viruses across the global oceans through metagenomic datasets from various large-scale sampling campaigns, such as Malaspina, Tara Oceans and Bio-GO-SHIP[14,65–69] (Supplementary Data 7).

Overall, the abundance of Fram Strait viruses peaked in the epipelagic (<200 m) around 60–70°N, with decreasing abundances towards 50°N, and typically no detection in subtropical and tropical epipelagic waters (Fig. 7a). A similar pattern occurred in southern hemisphere epipelagic waters, with peaks in abundance around 45–55°S (Fig. 7a). At individual vOTU level, 42% of vOTUs displayed bimodal peaks in abundance with separate maxima in each hemisphere (i.e., the two highest peaks in abundance occurred in separate hemispheres). The average peak latitude of vOTUs was 61°N and 51°S, respectively (Fig. 7b). Fram Strait viruses were also commonly detected in the deep ocean (here, operationally defined as sampling depth of >200 m). In particular, they were detected in 87% (338 of 390) of deep global samples. Like the epipelagic waters, deep-water viruses were more prevalent in northern and southern higher latitudes (Fig. 7b), though in deep samples the viruses were more commonly detected. Overall, however, when detected, their relative abundances were lower in deep samples than in shallow samples. Notably, there was significant variation in community structure of the detected Fram Strait viruses between epipelagic and deep samples (PERMANOVA: $F(1, 767) = 99.16$, $p = 0.001$; $R^2 = 0.1145$, 95 % CI [0.102 − 0.131])

which aligns with established differences in microbial communities in the deep vs. epipelagic ocean, and past results of virus biogeography[14]. Future work could consider the differences in the total virus communities between epipelagic and deep, as the current analysis utilizes only mapping to the Fram Strait viruses.

Expanding on these observations, we investigated the distributional patterns of the six Fram Strait modules on the global scale. All modules were more prevalent in the northern than in the southern hemisphere, but their relative distributions varied, with M1 and M2 being the most abundant in northern epipelagic waters, while M6 being more prominent in higher southern latitudes (Supplementary Fig. 8a), indicating a relatively restricted distribution of M1 and M2. In deeper waters, all modules were less prevalent than in epipelagic waters, with higher prevalence in the northern hemisphere. In deep waters, M6 was the most prevalent (Supplementary Fig. 8b).

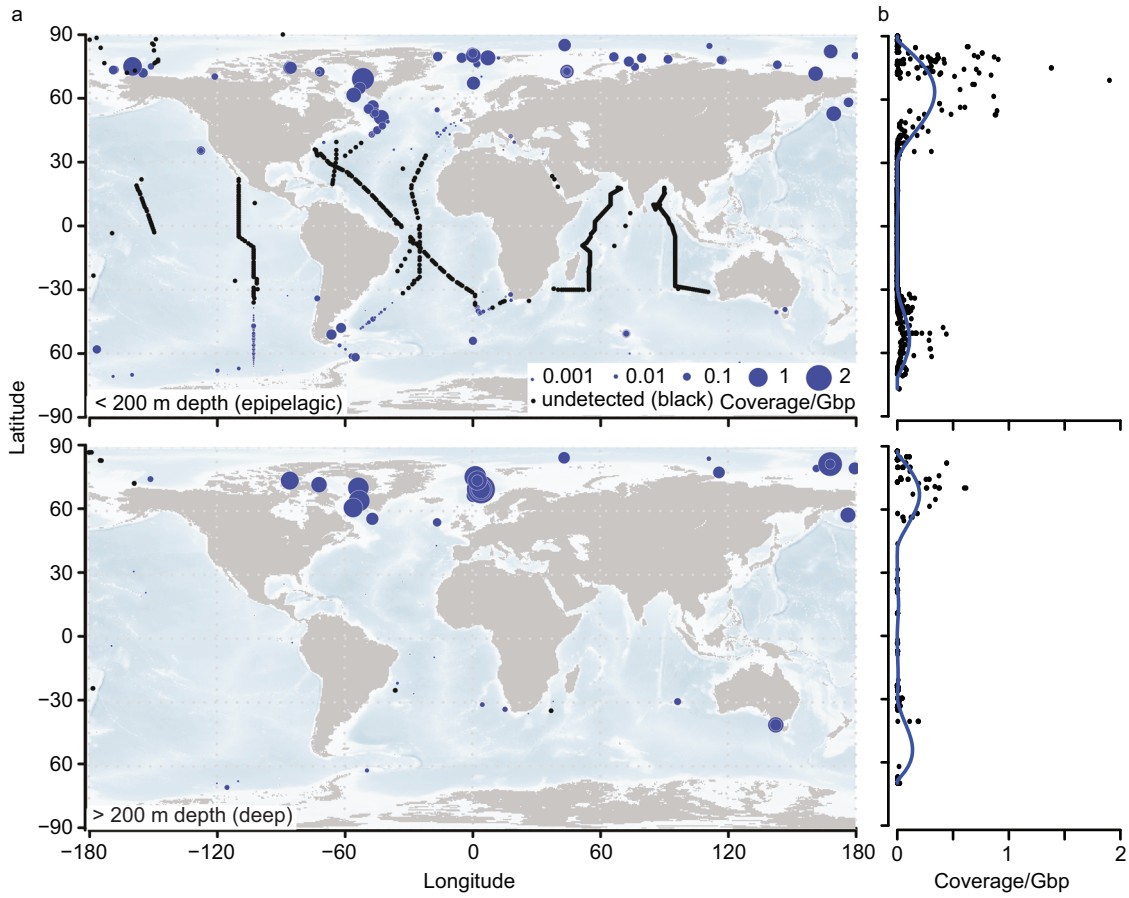

**Fig. 7 | Global distribution of Fram Strait vOTUs by comparison with short-read (0.2–3 µm size-fraction) metagenomes (Supplementary Data 7). a** Upper and lower map correspond to samples collected within the upper 200 m and deeper than 200 m, respectively. Abundance is calculated as coverage per gigabase pair of sequence (Coverage/Gbp). In the deep map, latitude and longitude were jittered to allow visualization of multiple depths at the same location. **b** Viral Coverage/Gbp plotted by latitude across all samples along with a Generalized Additive Model prediction, illustrating the trend of total viral Coverage/Gbp by latitude. For more information on samples, see Supplementary Data 1.

### Distinctive amino acid signatures of polar viruses

To expand on the global perspective, we assessed potential adaptations of viruses to polar waters by examining properties that have been attributed to cold adaptation in prokaryotes, in particular amino acid signatures that increase protein flexibility in cold environments. To do this, we examined the amino acid features of proteins from the Fram Strait viruses and from the GOV2.0 dataset, which spans both polar and non-polar sampling locations[14]. To ensure fair comparisons between the datasets, we focused only on GOV samples originating from ≤35 m water depth (i.e., the average depth of Fram Strait sampling) and first analysed the environmental context, which shows strong clustering of GOV2.0 polar samples to the Fram Strait, based on their similarity in distance to equator, oxygen, and temperature (Fig. 8a). Linking those environmental data to the examined protein features for all viruses, revealed that the aliphatic index and the nitrogen usage score were positively correlated with temperature (Fig. 8b), whereas other traits of potential cold adaptation, in particular polar charged and uncharged amino acids, showed significant correlation with one or more of the other environmental parameters, but not directly with temperature. This was also the case for *Caudoviricetes*, while *Megaviricetes* amino acid traits were more generally positively correlated with oxygen, temperature, and chlorophyll, and other vOTUs (non-*Caudoviricetes*, non-*Megaviricetes*) were generally positively correlated with salinity and oxygen (Supplementary Fig. 9). In the latter cases, these groups are more poorly sampled due to their low abundance resulting in weaker correlations overall, necessitating

further study. Examining the average values for these protein parameters according to sample and temperature, across all viruses, reflected the patterns observed via other statistical analyses (Supplementary Fig. 10), reinforcing our observations.

Beyond the amino acid level, we found that 17.0% of viral protein annotations were significantly enriched in high latitudes (253 of 1490), while another 7.1% were enriched in lower latitudes (106 of 1490) (Supplementary Data 8). Notably, among a variety of annotations enriched at high latitudes, most significant were chaperone proteins and stress response such as Cold (CSD) and Heat Shock Proteins (HSP70), DnaJ (Supplementary Fig. 11), as well as genes common in cyanophages in the lower latitudes including a photosystem gene (Photo_RC) and Transaldolase/Fructose-6-phosphate aldolase (TAL_FC)[70]. Together, our evidence suggests that biochemical and biological traits distinguish polar viruses from their tropical and subtropical counterparts, linked to amino acid sequences.

### Discussion

Harnessing long-read metagenomic data in high temporal resolution, we demonstrate a pronounced, annually repeating seasonality of viruses across multiple years in the Arctic Ocean. The seasonal viral dynamics correlated strongly with both the predominant prokaryotic host communities, as well as environmental parameters. We found that Fram Strait viruses and seasonal modules occur across cold waters of both hemispheres, while being mostly absent from subtropical and tropical waters.

a

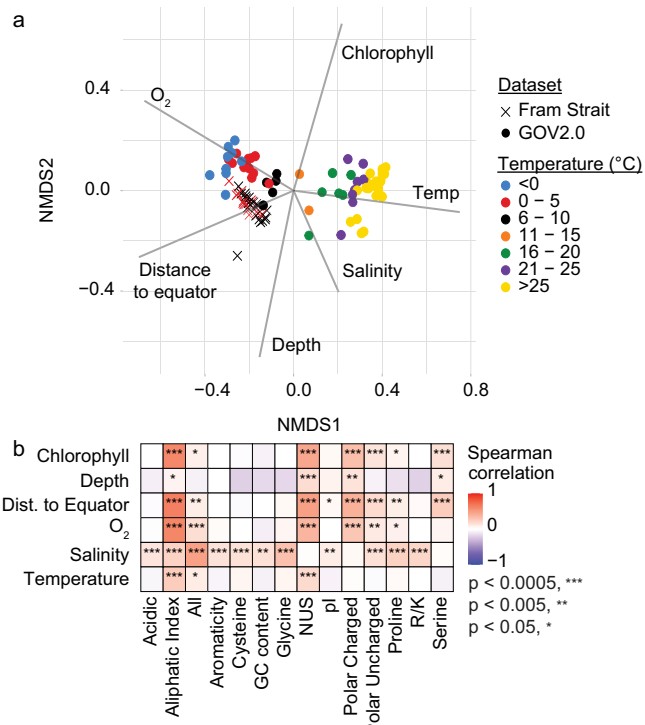

b

**Fig. 8 | Viral amino acid traits across environmental gradients. a** Nonmetric multidimensional scaling (NMDS) plot showing the environmental diversity of the sampled Fram Strait and GOV2.0 ecosystems. The Bray-Curtis distance similarity matrix was calculated based on the available environmental parameters of 38 Fram Strait and 69 GOV2.0 samples and used to generate NMDS coordinates of each sample. Point shapes represent dataset origin, and colors distinguish different temperature ranges. Vectors show correlations with environmental variables. This ordination provides context for the correlations to environmental parameters of amino acid traits shown in part b. **b** Heatmap plot illustrating the Spearman correlation coefficients of environmental parameters to amino acid traits, with *P* values represented by asterisks as indicated. The Spearman coefficients were calculated using a two-sided Mantel test using pairwise distances of each environmental parameter (Euclidean distance) vs. each amino acid trait (Bray-Curtis distance).

In terms of total virus prevalence, we found strong increases in the ratio of viruses to prokaryotes in the late summer months, mirroring findings from the Antarctic where viruses peaked in mid to late summer[23]. This increase was consistent regardless of the normalization used, namely either coverage-based virus to cell ratio (our main metric), or coverage per gigabase pair (which would also account for eukaryotic DNA). Our observation of strong virus seasonality in the cellular size-fraction contrasts with some of those of free viruses via fluorescence microscopy. This seeming contradiction may indicate a difference in the dynamics of infection and host-association, or loss factors of free-living viruses[71,72], for example photo-degradation of free viral particles[73,74]. Our results suggest that the number of viruses in the cellular size-fraction is in the same range (order of magnitude) as that of free viruses in seawater, and complement similar metagenomic marker-gene derived virus-to-prokaryote ratios which have focused so far on size-fractions that include free viruses[75]. This observation is consistent with estimated infection rates[10,76], especially considering that each infected cell can harbor a large number of viral genomes[77–80]. A further consideration is that our study focused on dsDNA viruses, thus not investigating the RNA virome[81–85]. Investigating RNA viruses, their seasonality and impacts on the prokaryotic community is an important future direction.

At the virus community level, the long-read and cellular size-fraction metagenomic approach likely aided assembly of the dominant viruses[86–88], which often suffer from assembly problems when short-

reads are utilized. However, we observed low similarity in opposing seasons compared to studies from more temperate areas[28–30]. The low similarity, reflective of low viral persistence in the ecosystem, may be due to the focus on host-associated viruses, thus excluding free-viruses with no prevalent host that could increase persistence (viral seed-bank hypothesis)[89,90]. Furthermore, the low persistence may also be in part related to the low frequency of predicted lysogenic viruses that we detected, which would otherwise be a potential persistence mechanism for viruses with rare hosts[91]. The very low similarity could also correspond to the relatively low coverage overall due to the long-read technology as well as dilution of reads from prokaryotes. Nevertheless, the strong seasonality reflects the strongly seasonal host communities[33,92].

The use of CCM in studying virus-host interactions presents challenges due to their rapid temporal dynamics. Viral latent periods range from hours to days, complicating interpretations based on monthly sampling intervals. The low overlap (3.7%) between CCM associations and iPHoP predictions may reflect limitations of our sampling resolution. Integrating multiple methodologies is critical for understanding the complexities of microbial interactions. Combining CCM findings with naive network analysis and iPHoP results provides a more comprehensive view of microbial community dynamics, with visual comparisons highlighting the biological significance of identified interactions. It should be considered that factors such as diversity and abundance can impact the total number of viruses detected in modules, and co-occurrence patterns overall. M5, for example, which has the lowest vOTU to prokaryotic ASV ratio, corresponds to the period during which vOTUs were the least relatively abundant. Additionally, during this period, vOTU richness was relatively high. The combination of these two factors challenges vOTU detection, and could result in vOTUs being excluded during pre-processing or missed altogether.

We observe a pronounced bi-modality in the latitudinal distribution of Fram Strait viruses, with peaks at high latitudes in both the southern and northern hemispheres. In particular, the peaks occur around the northern and southern polar fronts (70°N / 60°S)[93–96], suggesting that these boundaries of oceanic realms represent a hot-spot of polar-adapted viruses, possibly due to the pronounced biological and physicochemical gradients that are present, together with overall greater productivity. It might be, however, somewhat biased by a larger number of samples collected along these fronts. The high-latitude preference is also reflected within viral amino acid sequences, with signatures of cold adaptation that may contribute to their success in colder waters, thus expanding previous observations from the Southern Ocean[18] and polar eukaryotic viruses[97]. Further investigations to unravel the diversity and dynamics of viruses in Arctic and Antarctic are crucial, given that climate shifts are causing major biological and physicochemical perturbations in these regions. Additionally, in general, the detection of Fram Strait viruses in the deep ocean is notable, and potentially related to export from the surface and deep ocean currents, some of which originate at the Arctic. Further inspection of the specific vOTUs and other datasets at the global level could help further evaluate these ideas. Part of the explanation may be that vOTUs primarily observed in deep waters are from M6, the persistently present Fram Strait module; thus, they may be associated also with generally persistent/ubiquitous host lineages and/or have broad host ranges. Furthermore, the relative lack of seasonality in the deep sea may help with their consistent detection.

The peaks of Arctic viruses during summer and their specialization to high-latitude regions call for further process and targeted studies to elucidate the host communities that they interact with and influence. Previously, viruses of phytoplankton (cyanobacteria and eukaryotic phytoplankton) have been experimentally observed to exert less top-down pressure than eukaryotic grazers at high latitudes[98,99], but how this relates to the total microbial community

remains unknown. In any case, our study demonstrates that the impacts of Arctic viruses are likely very different depending on season and ecosystem state. Our results indicated a low percentage of predicted prophages without a seasonal enrichment, even in the cellular size-fraction studied. This suggests that the Piggyback-the-winner scenario is neither very prevalent, nor seasonally variable, though again this may be currently limited by computational predictions, and other biases like lytic viruses occurring within cells in more than one copy, and thus we may underestimate prophage prevalence. We found that most viruses are part of seasonally variable modules, and that often there is no similarity between opposing seasons (e.g., six months apart), at least at the depth sequenced at in the investigated cellular size-fraction. Hence rather than having a Constant-Diversity[46,47], Arctic virus communities are undergoing substantial seasonal change. Likewise, the prokaryotic community is also highly seasonal, thus the viruses are generally following and strongly correlated with host abundances. Thus, unlike how viral ecological models are sometimes idealized in a chemostat-like setting[100], viral communities' strong seasonality impacts how such models can be of use in the Arctic Ocean (such as in the Red-Queen hypothesis[29,48]). Arctic virus communities, with their strong seasonality where many taxa are not present year-round (or at least often not detected), may be impacted by how these dynamics play out elsewhere; likely, such dynamics occur primarily within seasons. Our analysis of microdiversity of Arctic virus communities demonstrated both instances of identical viruses across years, as well as a general tendency for vOTU to be underlaid by sequences with very high similarity but not exact matches to the vOTUs, akin to that seen elsewhere in the ocean[29,30,87], including the Arctic Ocean[14], though also some exact matches were observed across years. Thus, based on these results we cannot rule out either the Red-Queen hypothesis nor Constant-diversity. In each case, these dynamics could be further evaluated via more spatially-resolved temporal sampling.

In conclusion, our study advances the basis for understanding how viruses regulate and impact the dynamic and changing polar ecosystem, setting the stage for more detailed population dynamics studies, as well as process- and host-specific studies. Extended time-series observations will allow for improved understanding of the Arctic ecosystem impacts, in this ecosystem undergoing rapid change.

## Methods

### Seawater collection and eDNA sequencing
Research and sampling complied with relevant ethical and international regulations. Moored Remote Access Samplers (RAS; McLane) autonomously collected and fixed seawater from an average depth of 29 m in the eastern and western Fram Strait (Fig. 1) at weekly to fortnightly intervals between 2016–2020[42,58]. Sampling occurred in the framework of the FRAM / HAUSGARTEN Observatory. The resulting eDNA was used to sequence 16S rRNA gene fragments using primers 515F–926R[101], processed into ASVs using DADA2 as described under (https://github.com/matthiasswietz/FRAM_eDNA). Subsequently, we only considered ASVs with ≥3 reads in ≥3 samples, corresponding to a total of 3748 prokaryotic ASVs. To complement iPHoP host assignment predictions (below), we assigned taxonomy of prokaryotic ASVs via Greengenes2[102] using the classify-consensus-vsearch method[103] in the q2-feature-classifier[104] of QIIME 2[105].

DNA extracts from selected timepoints were additionally used to generate PacBio HiFi metagenomes at the Max Planck Genome Center, Cologne, Germany. Further details about the molecular analyses are described in previous reports[42,58]. In total, we herein analyze 94 amplicon samples and 47 metagenomes from the WSC, and 9 metagenomes from the EGC (Supplementary Data 1).

### Environmental parameters
Attached to the RAS were Seabird SBE37-ODO CTD sensors that measured temperature, depth, salinity, and oxygen concentration.

Sensor measurements were averaged over 4 h around each seawater sampling event. Physical sensors were manufacturer-calibrated and processed in accordance with https://epic.awi.de/id/eprint/43137. Employing multiple CTD sensors along the mooring depths enabled the determination of the minimum MLD at each sampling time point. Chlorophyll concentrations were measured via Wetlab Ecotriplet sensors. Surface water PAR data, with a 4 km grid resolution, was obtained from AQUA-MODIS (Level-3 mapped; SeaWiFS, NASA) and extracted in QGIS v3.14.16 (http://www.qgis.org). Polar water fraction, which is the proportion of polar water in the mixing of Atlantic and polar water masses, was calculated based on salinity and temperature[33]. Figure 1a map was created in QGIS v3.14.16 (http://www.qgis.org) using publicly available bathymetry data obtained from GEBCO. RAS illustration in Fig. 1a was generated via Inkscape (https://inkscape.org).

### Virus assembly, prediction, classification, and host prediction
Long-reads from each sample were assembled individually using hifiasm-meta v. 0.13-r308[106] with default settings. Viral sequences were predicted from both assembled contigs and the long-reads themselves using a combination of tools, based on a VirSorter2-based Standard Operating Procedure (SOP, https://www.protocols.io/view/viral-sequence-identification-sop-with-virsorter2-5qpvoyqebg4o/v3). First, in VirSorter2 v.2.2.3[107] we included all possible viral groups: dsDNA-phages, RNA viruses, ssDNA viruses, nucleocytoplasmic large DNA viruses (NCLDV), and Lavidaviridae. CheckV v.1.0.1[108] was then used to quality check the long-reads and contigs identified as viruses by VirSorter2[107] as well as to trim potential host regions from identified proviruses. Contigs identified as viruses by VirSorter2[107] (--min-score 0.5) with at least one viral gene (predicted by CheckV[108]) were further screened using DeepMicroClass v.1.0.3[109] (not included in the original SOP, but intended to further reduce false positives). Contigs that were classified as either eukaryotic or prokaryotic by DeepMicroClass[109] were discarded. Contigs that were identified as RNA viruses by VirSorter2[107] were not analyzed. We clustered the remaining viral contigs into vOTUs, as defined previously[110], using CD-HIT v.4.8.1[111,112] with the following parameters: cd-hit-est -M 100000 -c 0.95 -d 100 -g 1 -aS 0.85.

For classification, we used the *genomad annotate* function (default settings) of geNomad v.1.8.1[113] which uses taxon-specific marker proteins to assign taxonomy. To further resolve taxonomy to the subfamily level (Supplementary Data 2), we used vConTACT3 v.3.0.0b65 (https://bitbucket.org/MAVERICLab/vcontact3/src/master/). To predict hosts, we employed iPHoP v.1.3.3[52,114–117] with default settings, along with a custom database by adding previously assembled bacterial and archaeal MAGs from the Fram Strait (PRJEB67368) and removed MAGs from one oceanic study that was not manually curated, due to their high degree of potential virus contamination (PRJNA385857), which could result in false positives. When multiple host predictions were available, we selected the prediction with the highest confidence score.

### Mapping and normalization of viral abundance
Metagenomic long-reads from all samples were mapped to vOTUs (> 10 Kb, from assembly- and read-based approaches) using minimap2 v.2.28-r1209 (parameter: -x asm5)[118], and only the primary alignments were considered in subsequent analyses.

To quantify the abundances of vOTUs across metagenome samples, we employed a two-step normalization process to account for differences in read lengths and sequencing depth. First, we determined the mean coverage across the viral sequence length from mapped read counts (with a threshold that > 25% of the viral sequence must be covered). Second, we divided the mean coverage by the estimated number of cellular genomes in each metagenome, as predicted based on the mean coverage across 16 universal single-copy ribosomal proteins[119,120]. We call this metric coverage-based virus to cell ratio (cVCR). This approach is similar to that used to derive virus to

microbial ratios on virus to prokaryotic marker genes elsewhere[75], but considers the full viral coverage. In addition to this, we also calculated the coverage of each virus per gigabase pair (cVGB) of metagenome for a given sample and demonstrated that the two metrics yield comparable values (Supplementary Fig. 1). To gain an insight into the composition of the metagenome samples in terms of prokaryotic and eukaryotic reads, we employed Tiara v.1.0.3, a machine-learning tool, to assign domain-level classifications to the raw reads[121]. This revealed that the large majority of reads across the metagenomes, about 90%, were of prokaryotic origin.

## Estimation of viral diversity

Given that the vOTUs were derived from metagenomes with different sequencing depths, we employed an iterative subsampling approach to quantify and compare the alpha diversity of viruses across samples. For this, we applied 100 iterations of subsampling the vOTU count table at a range of different depths, spanning from 50 up to 30,000 counts at 50 count intervals. For this step, all vOTUs that exceeded the 25% breadth of coverage cutoff when using all the data for a given sample were utilized. During each iteration, richness, evenness and Shannon diversity were calculated, with mean values being determined for each 50-count interval. The functions *rrarefy, specnumber* and *diversity* from the *vegan* package[122] were used to perform subsampling and alpha diversity calculations. The mean values were used to generate a rarefaction curve of alpha diversity using the *ggplot2* package[123]. To assess shifts in diversity over time, we compared the mean richness values obtained from subsampling at the 1000 count interval. In addition, we computed and compared the slopes of the sample rarefaction curves up until the subsampled depths, which represents the rate of vOTU discovery with increasing viral read counts. Comparing the slopes across samples enabled an assessment of whether the sample richness patterns observed at the subsampled depth would change if the viral read count was increased. To further explore this, we also employed the iNEXT v.3.0.1 package[124,125] to estimate the vOTU richness based on extrapolations to 30,000 viral read counts.

## Co-occurrence network

To identify temporal co-occurrence patterns, cVCR values per vOTU and relative abundance of prokaryotic ASVs were converted into temporal profiles by Fourier transformation. Temporal profiles were constructed based on 16 Fourier coefficients, which capture the majority of observed vOTU and ASV peaks within the four-year period. Pairwise correlations between individual temporal profiles were then computed between all vOTUs and ASVs. Higher Pearson correlation values indicated similar temporal profiles. For network construction, we first calculated Pearson correlations for all pairs resulting in an undirected graph, from which we only considered correlations > 0.7 after multiple testing corrections using the Benjamini-Hochberg procedure. This threshold was determined based on previous findings on similar data[49,58] as well as preliminary analyses conducted on our dataset, which indicated that interactions exhibiting a correlation below this value often lacked biological relevance. To delineate strongly connected components representing co-occurring taxa, the Louvain community detection algorithm was applied to the entire graph[126]. Next, the putative associations were further evaluated based on Convergent Cross Mapping (CCM) in order to discern causal relationships between taxa in time-series data, as outlined by ref. 127. CCM enables the prediction of a species' time-series based on the knowledge of another species' time-series. Given the nature of lytic virus-host systems, where interactions can be short-lived, we acknowledge that this method must be carefully interpreted within the context of our sampling strategy (i.e., biweekly to monthly on average). Initially, we constructed a CCM network encompassing all pairwise combinations. Subsequently, we extracted the in- and outgoing edges between

nodes that were also connected in the co-occurrence network, utilizing resources from (https://gitlab.com/qtb-hhu/marine/publications/framphages2024). To quantify the strength of relationships, we employed Normalized Mutual Information (NMI) to account for non-linear relations[49]. A permutation approach was employed to compute significance values for edge weights, with the objective of determining whether the NMI values exceeded those expected for random edges. Further details on the construction and validation of the CCM network can be found in ref. 49. The CCM network was visualized in Cytoscape v.3.10.1[128]. Additionally, to identify potential time-lagged correlations, we employed extended local similarity analysis[129] with a maximum delay of two sampling points on the original non-Fourier transformed data.

## Statistical analyses

Bray-Curtis similarity, Mantel tests, CCA, and Spearman correlation were performed in R v. 4.3.1[130] using the *vegan*[122] and *Hmisc*[131] packages. We included only vOTUs with a breadth of coverage > 0.25.

## Cyanophage phylogenetics and comparative genomics

Putative cyanophages were predicted via VPF-class v.0.1.2[56] and iPHoP[52] host predictions. Subsequently, we focused on the most confident predictions by selecting cyanophages harboring the *psbA* core photosystem, identified by hmmscan[132] of viral predicted proteins (prodigal[133]) PFAM for Photo_RC (PF00124). Viral and prokaryotic reference *psbA* were extracted from the vConTACT2 viral reference database (ViralRefSeq-prokaryotes-v211[53]) and GTDB prokaryotic reference database (version 214[134]) by hmmscan[132] with PFAM Photo_RC. Eukaryotic reference *psbA* genes were extracted via Uniprot (gene_exact:psbA) AND (taxonomy_id:2759, reviewed Swiss-Prot, $n = 157$)[135]. A preliminary tree was constructed from these sequences to differentiate between D1 (*psbA*) and D2 (*psbD*) photosystem II (PSII) reaction center proteins, by alignment using MAFFT[136] followed by trimming with trimAl (-gt 0.2)[137], and FastTree[138] for phylogenetic reconstruction. *psbD* were identified manually and excluded. The remaining sequences were considered, and phylogenetic reconstruction performed a second time with the same alignment and trimming steps (but with -gt 0.8); at this stage, tree building was performed with IQ-TREE (settings: -B 1000, -alrt 1000, -m MFP)[139].

## Mapping to global metagenomes

Metagenomes from ten oceanic datasets[14,65–69,140–145], including Tara Oceans, Malaspina, and Bio-GO-SHIP (Supplementary Data 7), were downloaded from NCBI, and quality-filtered with Trimmomatic (parameters: LEADING:3 TRAILING:3 SLIDINGWINDOW:50:30 MINLEN:50)[146]. All data utilized is previously published. Only DNA samples from discrete depths (e.g., not composites of the mixed layers or similar) and the prokaryotic size-fraction (0.2–3 μm) were used to allow comparison with our dataset. Multiple other size fractions are available, but the 0.2–3 μm was utilized because of our focus on prokaryotic dsDNA viruses, its comparability to the present Fram Strait virus dataset, and because it is among the most common range sampled (i.e., many samples available). Reads were mapped to the clustered contigs using bbmap.sh from the BBTools package (version 39.01, parameters: minid=95 idfilt=95) (https://sourceforge.net/projects/bbmap/). Trimmed mean coverage was calculated via CoverM[147] on contigs with minimum covered fraction of > 25% for each sample. Trimmed mean coverage values were then divided by total Gbp per quality filtered paired reads. Maps were generated via publically available NOAA bathymetry data via getNOAA.bathy and the *maps* package in R. For determining abundance trends in the total dataset and at the module level, we computed a Generalized Additive Module (GAM) between cVCR and latitude using the *gam* function from the *mgcv* package[148], with ten degrees of freedom (k = 10). For plotting the predicted GAM, negative values were converted to 0. For

calculating the maximum latitude that viruses occurred in each hemisphere, we used the GAM model, computing where peaks occur and then determining which two peaks had the highest abundance. For evaluating statistical differences in community similarity between epipelagic (< 200 m) and deep (> 200 m) samples, we first removed samples that had less than 0.0001 cVGB (n = 769 samples remaining, including 435 and 334, shallow and deep samples, respectively) and vOTUs that had less than 0.0001 summed cVGB across all samples (n = 4,693 vOTUs remaining). Then, we ran adonis2[122,149] with 999 permutations based on the epipelagic and deep grouping, with 95% confidence intervals for the $R^2$ statistic estimated using 1000 bootstrap replicates.

### Microdiversity of viral contigs

To examine microdiversity among vOTUs, we used two separate approaches. First, to examine the extent that exact, long virus sequences re-occur across years, we used CD-HIT v.4.8.1[111] to cluster viral contigs > 10 Kb at a 100% sequence identity threshold (cd-hit-est -M 100000 -c 0.95 -d 100 -g 1 -aS 0.85). We then identified clusters that appeared across more than one year throughout the four-year sampling period. Second, we used inStrain v.1.9.0[150], which compares reads mapped to vOTUs to characterize virus populations based on nucleotide diversity, total counts of single nucleotide substitution (SNS) and single nucleotide variation (SNV), and other related metrics. We applied a 98% minimum ANI, the most stringent threshold recommended by the developers, and used a minimum coverage of five to call a variant. To confirm a SNV, we required a minimum SNV frequency of 0.05 and a false discovery rate of 1e-06. Additionally, we filtered for contigs where more than 25% of their bases were covered by at least one read, i.e., breadth > 0.25. Only vOTUs that exceeded the coverage minima more than three times throughout the sampling period were included in the analysis.

### Calculation of amino acid traits

Proteins were predicted from GOV2.0 and Fram Strait viruses via Prodigal v.2.6.3 (parameter: -p meta)[133]. Then, amino acid composition and various biochemical properties of the predicted protein sequences were assessed using custom R scripts available at https://github.com/alyzzabc/fram_strait_viruses_2016-2020. Briefly, protein sequences containing the ambiguous amino acid "X" were removed, followed by counting the occurrences of each proteinogenic amino acid. The frequencies of single amino acids (G, S, P, C) and of amino acid groups (acidic, polar uncharged, polar charged, aromatic) were calculated relative to the protein length. For the amino acids arginine (R) and lysine (K), the ratio R/K was calculated instead of occurrence. The molecular weight was calculated according to[151] and for calculation of the aliphatic index we followed[152]. The pI function from the Peptides package[153] was used to calculate the pI of the predicted proteins using the EMBOSS scale. The nitrogen usage score (NUS) was calculated using the following formula:

$$NUS = \frac{4F(R) + 3F(H) + 2F(KNQW) + F(DESTGPCAVILMFY)}{n(protein)} \quad (1)$$

where F(X) corresponds to the absolute count of each amino acid and n(protein) to length of the protein.

To analyze the variation of the viral protein features in relation to the environmental conditions, we omitted samples collected from a water depth > 35 m followed by calculation of the Bray-Curtis distance similarity matrix and generation of NMDS coordinates using the metaMDS function from the vegan package[122].

To calculate pairwise correlations, the vegan package[122] was used: the distances for each protein feature (Bray-Curtis) and environmental condition (Euclidean) were calculated using the vegdist and dist function, respectively. Pairwise Mantel tests of each environmental parameter vs. each amino acid trait were done using the Spearman method with 9999 permutations via the mantel function. The NMDS plot and the correlation heatmap were prepared using the ggplot2 package[123].

### Calculation of protein family enrichments by latitude

For determination of proteins enriched in polar regions, we used the prodigal-predicted proteins for GOV2.0 and the Fram Strait virus dataset. These predicted proteins were annotated with pfam[154] via hmmer[132] using the protein-specific cutoffs for significant (cut_ga). Then, individual protein families were summed within a given sample. Proteins that had prevalences of less than 10% were removed (n = 1,490 remaining). Then, ANCOM-BC[155] was used to determine which annotations were enriched at latitudes greater than 60°N or S, or less than 60°N or S.

### Reporting summary

Further information on research design is available in the Nature Portfolio Reporting Summary linked to this article.

## Data availability

Raw metagenomic reads are available under ENA BioProjects PRJEB67368 (WSC) and PRJEB52171 (EGC). 16S rRNA amplicon reads are available under PRJEB43890 (2016-2017), PRJEB43889 (2017-2018), PRJEB67813 (2018–2019), and PRJEB66202 (2019–2020). Physicochemical parameters are available at PANGAEA under (https://doi.org/10.1594/PANGAEA.904565) (2016–2017), (https://doi.org/10.1594/PANGAEA.904534) (2017–2018), (https://doi.org/10.1594/PANGAEA.941126) (2018–2019), and https://doi.org/10.1594/PANGAEA.946508 (2019–2020). vOTU contigs and network files are available via figshare DOIs (https://doi.org/10.6084/m9.figshare.28045856) and (https://doi.org/10.6084/m9.figshare.28045826,) respectively. The psbA alignment, trimmed alignment, treefile, and labels for cyanophage references are available via figshare (https://doi.org/10.6084/m9.figshare.28398173).

## Code availability

Code and inputs for reproducing workflows and figures are available via GitHub https://github.com/alyzzabc/fram_strait_viruses_2016-2020/ and https://doi.org/10.5281/zenodo.15490229. All scripts to reproduce the CON and CCM networks are available via gitlab https://gitlab.com/qtb-hhu/marine/publications/framphages2024. CON and CCM calculations can also be performed using the tool OTTER, available as a GUI (https://doi.org/10.5281/zenodo.13840702).

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

## Acknowledgements

Christina Bienhold, Katja Metfies, Ian Salter and Antje Boetius co-designed the mooring strategy and coordinated sample collection and processing. Katja Metfies coordinated amplicon sequencing. Wilken-Jon von Appen, Sinhué Torres-Valdés, and Daniel Scholz carried out physicochemical and oceanographic measurements. We thank Jana Bäger, Theresa Hargesheimer, Jakob Barz, Anja Batzke, Rafael Stiens, and Lili Hufnagel for RAS operations; Normen Lochthofen, Janine Ludszuweit, Lennard Frommhold and Jonas Hagemann for mooring operations; Jakob Barz, Swantje Ziemann and Anja Batzke for DNA extraction, amplicon library preparation and sequencing; and Bruno Huettel, Christian Woehle and the technicians at the Max Planck Genome Center in Cologne for metagenome sequencing. Captains, crew and scientists of RV Polarstern cruises PS99.2, PS107, PS114, PS121 and PS126 are gratefully acknowledged. This project has received funding from Polarstern grants AWI_PS99_00, AWI_PS107_05, AWI_PS114_01, AWI_PS121_07, AWI_PS126_05, and AWI_PS126_07. Further support came from the Helmholtz Association, the Max Planck Society, and a Helmholtz Young Investigator Grant to D.M.N. M.W. was supported by the German Research Foundation grant 522416631 within the SPP 1158.

## Author contributions

A.M.C. identified, annotated, mapped, and assembled viruses. A.M.C. and D.M.N. analyzed viral and prokaryotic community data. T.P. contributed normalization techniques and alpha diversity metrics. E.O. and O.P. performed network analysis and module identification. J.M. performed amino acid analyses. M.W. and D.M.N. instigated the study. D.M.N. designed and coordinated the analyses. A.M.C. and D.M.N. wrote the paper with input from all co-authors, especially T.P. and M.W.

## Funding

## Competing interests

The authors declare no competing interests.
