## [Transparent Peer Review file · Nature Communications]

Arctic Ocean virus communities and their seasonality, bipolarity, and prokaryotic associations

Corresponding Author: Professor David Needham

Version 0:

Reviewer comments:

Reviewer #1

(Remarks to the Author)

Calayag et al. present a well-conducted and comprehensive study of dsDNA viral ecology in the Fram Strait, Arctic, addressing a crucial gap in our understanding of virus dynamics in polar regions. The study demonstrates the seasonal structuring of viral communities over a four-year period, with distinct annual peaks in the virus-to-prokaryote ratio during the summer months. Additionally, the discovery that a significant proportion of Arctic viruses exhibit bipolar distributions further contributes to our knowledge of their biogeography. The use of convergent cross mapping network to identify virus-host interactions within ecological modules enhances our understanding of DNA viruses' ecological roles and interactions. Moreover, the analysis of amino acid signatures in viral proteins provides valuable insights into how viruses adapt to such extreme environments.

Overall, this study is valuable to environmental virologists, offering both important analysis results and data on the ecology of viruses in polar environments. The use of long-read metagenomics strengthens the findings by capturing larger genomic fragments. Additionally, the use of cVPR and CCM provides novel methods for understanding virus-host interactions. However, before this manuscript is considered for publication, I have some suggestions and comments listed below.

General Comments:

1. In Fig. 5b, many vOTUs do not encode structural or packaging-related genes, raising my concerns about the completeness of these cyanophage genomes. I wonder if this is an issue of labeling in this plot or if it reflects the nature of these genomes. It would be helpful if the authors could provide CheckV results along with basic statistics for the viral genomes to clarify genome completeness and quality.

Additionally, the bootstrap support in Fig. 5a for the basal topology appears low, with most nodes having bootstrap values < 80. To improve the reliability of this cyanophage phylogeny, I suggest the authors consider using alternative genes or incorporating cellular psbA genes into the tree.

2. While Convergent Cross Mapping (CCM) appears to be a robust method for detecting causality in complex systems, I have some concerns about the application to lytic virus-host systems (high proportion in this study), where the short duration of interactions (with latent periods ranging from few hours to days) may complicate the inference of directional causality under a resolution of monthly sampling. This could potentially explain the limited overlap (4.3%; LN 221) between CCM associations and iPHoP predictions. I recommend that the authors provide further justification for their choice to use CCM in this context, potentially addressing these challenges in the methods section or adding a caveat in the discussion.

Additionally, the use of certain parameters, such as a corrected Pearson correlation threshold (>0.7), may overlook biologically meaningful interactions, as viruses and their hosts do not always perfectly co-occur and gain a high correlation coefficient. A more detailed explanation of these parameter choices would be helpful to people who want to follow the study. Alternatively, the authors could present a naive network (without edge filtration) to compare against their CCM-based inferences, particularly regarding module clustering and mutual pairs identified by iPHoP.

3. The amino acid composition analysis is presented as pooled data across all viral genomes, making it difficult for me to determine whether the observed amino acid signatures are driven by environmental adaptation or reflect community composition differences, especially given the strong viral endemism in polar regions. I suggest that authors consider analyzing amino acid traits in relation to phylogenetic distance, or constraining the analysis by same viral taxonomic groups, to better distinguish evolutionary lineage from environmental adaptation.

4. Authors have highlighted amino acid traits, but it would also be valuable to consider the role of other mechanisms, such as gene gain, in viral adaptation. Have authors examined whether there are gene orthologs specifically present or absent in polar regions? Or an additional discussion on how mechanisms may influence viral adaptations in polar environments could help readers understand virus adaptation to environments.

Minor Comments:

LN135-139: The color palette in Fig. 2 makes it challenging to check the left panel. It would be helpful to use two distinct color sets between the two groups (Jan-May vs June-Sep).

LN142: The lower diversity and slower accumulation observed in summer could be a result of higher overall biomass during that season. Few dominant bacteria and viruses may account for the majority of reads at the current sequencing depth (10Gb), so the detection of rare viruses has become relatively difficult. Please consider adding this as a potential limitation in the text.

Fig. 3b: Using different shapes for the dots to represent different years could make it easier to understand temporal trends across four years.

LN168: I'm uncertain whether the phrase "shaped by their hosts" is appropriate, as it's difficult to attribute the observed viral patterns to direct host influence, given the potential roles of other prokaryotes. Additionally, I am a bit confused about which pair in Supplementary Data 1 represents the structure of viruses in the main text. Authors could add the values for each variable in the main text.

LN179: Please add the number of 'following' and 'caused by'.

LN192-193 & Fig. 4d: The ratio of vOTUs/ASVs in M5 is significantly lower than in other modules, and the value is somehow counterintuitive. Could this be due to that most winter viral nodes being discarded during network preprocessing? A bit further discussion on potential reasons behind this observation would be helpful.

LN265: Are the 58 vOTUs mentioned evolutionarily related (similar taxa)?

Fig. 4C: Please ensure the p-values have been corrected for multiple testing.

Supplementary Data: It is not easy to understand the headers in supplementary data 1 and 2. Please consider using one spreadsheet to include descriptions of the column headers for all tables.

(Remarks on code availability)

Reviewer #2

(Remarks to the Author)

Review Nature Communications. NCOMMS-24-55589

Arctic Ocean virus communities: seasonality, bipolarity, and prokaryotic Interactions

This study investigates the seasonality, abundance, diversity, associations with prokaryotic hosts, and environmental factors of Arctic viruses in the Fram Strait over a four-years period. The authors used 47 long-read metagenomes obtained from the cellular size-fraction, which were automatically collected. It was found that viruses exhibited annual peaks, and that the composition of the viral community was strongly correlated with prokaryotic community composition. Additionally, it was analyzed presence of the polar virus in metagenome from different latitudes, concluding that virus abundance is higher in high-latitude regions of both poles, highlighting adaptations for cold environments.

First, I would like to positively emphasize the rigorous mathematical and statistical methods employed by the authors in this work. While seasonal cyclic variability in viral communities has been observed in prior studies, the extended sampling period here provides a more robust dataset than previous research. The constructed metagenomes alone represent a valuable resource database. However, some sections lack sufficient biological interpretation and related discussion. Further clarification from the authors on the following specific points is recommended to enhance the overall quality of the manuscript.

In general, more quantitative data should be included in the main text and as Supplementary Data. Table of Supplementary Data 4, in particular, could be expanded to include specific information such as the number of viral sequences in each collection, the depth (or range) of the water column from which each sample was obtained, the maximum length of the viral raw reads obtained (L239), the estimate of the number of prokaryotic genomes in each metagenome, and, if possible, parameters indicating viral (L142) and prokaryotic diversity. Additionally, a global overview of prokaryotic taxonomy retrieved by ASVs and their periodicity is needed in the Supplementary Data to improve the interpretation of the seasonality observed in the viruses (L142, L80). Details regarding the sizes and number or replicas of the 16S rRNA libraries should also be provided as Supplementary Data). Furthermore, data on the ASV-vOTUs involved in the six modules, as well as those identified by iPHoP (L181 & L214), must also be included.

To better grasp the magnitudes involved (as the authors provide statistical data but almost no "raw numbers"), it would be helpful to include precise information in the main text. For example, it would be useful to know how many viral sequences were identified among the nearly 200 000 sequences obtained per metagenome (L96); how many viral sequences represent the 5662 vOTUs identified by the authors; which vOTU was the most abundant and how it varied along time and how many vOTUS are involved in each module (L173-195). Additionally, the 3748 ASVs mentioned in L391 should be cited earlier, at the beginning of the Results sections.

Specific questions:

1.-In my opinion, the title or the article is inadequate. First, not all viruses form the community were considered, as RNA viruses were excluded from the analysis (L422), and the authors focused primarily on dsDNA viruses. Importantly, the authors examined viruses found exclusively in the cellular fraction; i.e., those actively infecting cells at the sampling points or

integrated into cellular genomes as lysogens. Additionally, there is limited information on prokaryotic “interactions” per se, as host identification relies exclusively on computational predictions. I believe a title emphasizing the seasonality and the dynamics of the viral community would better reflect the study’s scope and attract a broader audience. Moreover, only viruses present in the cellular fraction are considered in the analysis, rather than the complete community- an important detail that should be more clearly stated throughout the manuscript.

2.-The introduction provides quotes related to the ‘Arctic Atlantification’, but this information is not directly relevant to the goals or findings of this manuscript. In addition to the hypothesis proposed by the authors, L79, which is not entirely novel (reference 22, 25), I would encourage the authors to present hypotheses in the introduction regarding how polar viral communities would fit in the ecological models already described for virus-host populations., e.g., ‘Piggyback-the-Winner’ theory, Red-Queen dynamics, Constant Diversity Model, etc. (F.R. Valera et al. 2009, B. Knowles et al. 2016, F. H. Coutinho et al. 2017, H. Alrasheed et al. 2019, J.C. Ignacio-Espinoza et al. 2020). This would help frame the discussion to better align with fit the Lotka-Volterra dynamics observed.

3. There are two points that need to be clarified regarding the methodology used:

(i) Importantly, the authors claim that the lytic lifestyle was predominant over lysogeny. Although I find value in the analysis conducted by the authors, I have a primary concern regarding the number of lysogens detected. Beyond bioinformatic predictions, the authors lack direct evidence to fully support this conclusion. Many lysogens do not have recognizable integrases, which is a general challenge that needs to be addressed (out in the context of this work). In my view, viruses replicating inside the cells (i.e., lytic viruses) would outnumber lysogens in term of copies within the cellular metagenome, making them easier to assemble with the long-reads used, compared to lysogens. Therefore, a strong bias could obscure the results.

(ii) Cut-off used to define the abundances of vOTUs. The authors use only 25% of the viral genome length, whereas typical values range from 50 to 70% at a minimum.

4. The authors focus on Cyanobacteria and the dynamics of putative cyanophages (L232-257) due to their relevance to changing conditions in Arctic ecosystems (but it is unknown in which module are founded), along with another nice example of the Nitrososphaerales. In my opinion, there is a lack of detailed information regarding the largest group, which is represented in several modules and with multiple identified virus-host pairs, the Pelagibacteraceae. Providing information on how the Pelagibacteraceae ASVs are distributed across the different modules, along with the assignment vOTUs, would better complement this section of the manuscript.

Other details:

Just as a suggestion, did the authors consider whether Supplementary Figure 3c could appear as a main figure (perhaps in place of Figure 4a)?

L63. Please, clarify. Sandaa et al 2018, (reference 22) found that (cited textually) “Viral diversity and virus-to-prokaryote ratios (VPRs) dropped sharply at the commencement of the spring bloom but increased across the season, ultimately achieving the highest levels during the winter season. These findings suggest that viral lysis may be an important process during the polar winter, when productivity is low”. Therefore, this contradicts what is stated in L59, 60.

L93-94. Did the authors retrieve eukaryotic DNA in their sequencing? Since no pre-filtering was applied, there is a high possibility of this occurring, yet it is not mentioned in the manuscript. If present, could this affect the cVPR calculations?

L98-99. The program CheckV is mainly used to confirm the nature of the viral sequences, not to make predictions per se. Therefore, remove from this form the line, and then the two approaches mentioned in L99 that refers the authors, will be correct.

L107. “95% sequence clusters”. Please modify to indicate that this refers to 95% ID.

L109. Please note that it is recommended to follow the updated nomenclature for phages. The bacteriophage families Myoviridae, Podoviridae, and Siphoviridae have been reclassified under a new taxonomic system (International Committee on Taxonomy of Viruses (ICTV), 2021). These are now part of the order Caudoviricetes, which encompasses tailed bacteriophages. Myoviridae members are often reclassified in families like Myoviricetes, while Podoviridae members generally fall under Autographiviridae, and Siphoviridae have been split across families like Drexelviridae and Demereciviridae, among others.

L111. Phycodnaviridae continues to be cited in L122, L199 and Figure 3, so please, update this sentence accordingly.

L166. Please define ‘PAR’ as it is the first time it appears in the text. Also, what parameter exactly is the “polar water fraction” in Figure 3b? Additionally, “iceDist” and “icePast” are not mentioned in the manuscript (also in Figure 4).

L287. Is this in disagreement with the statement in L280? First, authors must clarify whether the vOTUs detected in surface waters are different from those detected in deep waters. If Fram Strait viruses are also abundant in deep ocean samples, does this imply that similar hosts are more abundant at these depths? Additionally, which depth range is being referred to? This information needs to be either depicted in the figure or stated in the Supplementary Data 3. It is generally understood that deep and surface prokaryotic communities differ... then, could mixed waters be influencing this observation (although the authors specifically mention that these “mixed” samples were not considered in the analysis)? The authors should provide a more detailed discussion on this point (L291-298).

L310-L317. Please, provide more details on the changes observed in the amino acid composition of the viruses from Fram and GOV2.0 datasets versus those found in in northern or southern lower latitudes. The authors provide statistics, but there is no percentage or other parameter that can be quickly appreciated.

L331. The quantity of free viruses cited here were obtained in 2014 during in partial transect compared to the one followed in this work. Although due to the periodicity observed, it is likely that those numbers were similar, the authors could support this by using metaviromes and metagenomes recovered from the same sample and check the vOTUs abundance recovered in both.

L438. RaFAH results are not shown anywhere in the text. The iPHoP results incorporate RaFAH, which helps improve

phage-host association predictions. Please, correct the sentence.

L452. Is reference 88 correctly placed here? Please cite the original source where it is described.

L536. The metagenomes chosen to map the viruses were those obtained using a prokaryotic size fraction of 0.2-3 microns. However, authors do not cite to use any pre-filtering, so, collections with larger sizes could also be suitable. Please, revise the sentence accordingly, as these metagenomes were selected because the authors focused on prokaryotic dsDNA virus relationships.

Figure 4a. Please, modify this figure to better differentiate viruses from ASVs.

Figure 5a. *psbA* gene in italics.

Please, enlarge the figures for improved readability. At their current size, it is challenging to distinguish finer details.

Figure S3c. May the authors order the microbial family names accordingly to a taxonomic relationship? It would make more sense than organizing them alphabetically.

(Remarks on code availability)

I was able to download and run the code. They all seemed to work properly. However, I could not check the complete results as I do not have the appropriate inputs available.

Reviewer #3

(Remarks to the Author)

The authors present a well-written paper about a novel dataset that describes a rare 4-year time series of viral metagenomes from the Arctic. They introduce the broad seasonal patterns of viruses they observed and add extra analysis by comparing the viruses they found with other publicly available datasets of that type. They use a fairly straight forward analysis with appropriate and common pipelines and tools.

However, there is one noticeable exception: the authors look at a mix of morphotype classification (podo-, myo- and siphoviridae) for bacteriophages and genetic based taxonomic groups for eukaryote viruses (Phycodnaviridae, Mimiviridae). This seems like an outdated approach since the ICTV reported that they are not monophyletic, phylogenetically solid groups (See e.g. "Taxonomy of prokaryotic viruses: 2018-2019 update from the ICTV Bacterial and Archaeal Viruses Subcommittee"). A direct comparison between those different types of classification groups not ideal, and there is no clear reason why the more recent classification has not been chosen. The main focus of the paper is not on that aspect, so it is not an unfixable problem. But please update the analysis according to the most recent ICTV guidelines, or at least justify why the chosen approach is still valid (I will leave that up to the editor). The authors should at minimum acknowledge this issue in the text and explain why this comparison was chosen, rather than the more up-to-date classification established in recent years. It is crucial that the authors carefully go through the entire manuscript and make absolutely sure that they provide a citation for every software tool, package and database and whatever else they use. Studies like this one would not be possible without all the public databases and free software tools developed by other scientists, it is only good practice to give everyone credit for their work.

Minor comments:

22-23: Perhaps specify that the paper is about viruses of microbes.

26-28: Wording could be clearer.

62-63: A new paper has been released after the authors would have submitted their manuscript, but there is a viral metagenomic time-series that might be relevant to consider here. <https://academic.oup.com/ismej/advance-article/doi/10.1093/ismejo/wrae216/7833430>

78-80: It is known that microbes in the polar regions are highly seasonal and, obviously, virus abundance is directly linked to host abundance, a metric the authors use as well. Given that seasonality of viral communities has been shown in other regions and has been speculated for polar viruses as well, this is not a good hypothesis in my opinion. The authors present a very important and interesting dataset, which as far as I know is the first virome time series from the Arctic Ocean. They should emphasize the novelty of this dataset, rather than the general concept of viral seasonality.

An additional side point: the manuscript only provides data from the Fram Strait, so the hypothesis should be about Arctic seasonality, not polar seasonality, in any case. It's reasonable to speculate similar patterns for the Southern Ocean, but the seasonal dataset only really covers the Arctic.

90-91: What was the depths these samples were collected from? Please clarify.

117: s missing for ratios?

166: Please introduce the meaning of abbreviation PAR.

162-171: Please provide a value or a figure for this. The stats are buried in a difficult-to-read supplementary table, but this is a fairly important point and the following paragraph is building on that point. It should be made clear to the reader how significant that correlation is compared to oxygen and MLD. The authors provide a main figure (3) and a percentage to show the results for the less important physiochemical drivers, but not for the correlation with their hosts. The text and figures should reflect the more important aspects of results first.

241-242: This is not surprising since the authors group the viruses based on the outdated morphological classification, which is polyphyletic.

287: Define what depth is considered deep ocean. Based on context, > 200m depth.

467: Please cite the authors of the *vegan* and *ggplot2* packages and acknowledge the use of R.

501: Please reference the original Cytoscape paper.

518: Please cite the original vConTACT2 paper.

519: Please cite the *hmmer* paper.

520: Please add citation for UniRef.

532: Please add citations for Tara Oceans, Malaspina Expedition and Bio-GO-SHIP. They are hidden in a supplementary

table as a link only, but should be properly cited in the text itself too. I will stop checking if the authors have referenced every tool, database and package, since they tend to only add a citation upon first mention, if at all, so it's tedious and easy to miss.

(Remarks on code availability)

The codes seem fine, there is enough instructions and explanations.

Version 1:

Reviewer comments:

Reviewer #1

(Remarks to the Author)

I'd like to thank the authors for their efforts in addressing my comments and improving the manuscript. After reading the rebuttal letter and the revised manuscript, I find that most of the responses are adequate, and the revisions are well-explained and feasible. The improvements, such as clarifying genome completeness and expanding discussions on arctic adaptations at gene level, are appreciated. Overall, the manuscript has largely improved.

I still have a few remaining suggestions to enhance robustness, transparency, and clarity of this paper.

1. First of all, I have to say sorry that I made a mistake in my first revision about the total sequencing depth. I originally thought it was 10 Gb per sample, but, in fact, it is only 1 Gb (196,489 reads × 5,435 bp per read). This low sequencing depth makes the quantification of viruses more challenging than I expected, especially given that viral diversity does not reach saturation in the dataset.

The updated results indicate that viral diversity peaks in late summer to autumn, which is the somehow opposite of what was stated in the initial submission. This is a big shift in the overall viral seasonal dynamics.

My question is:

The error bars in Figure 2B are very large, making it difficult to compare diversity across months with confidence. Would it be possible to restrict the analysis to only samples with adequate viral reads when calculating diversity? Alternatively, it might be helpful to present a fully expanded diversity/richness plot for each year, similar to what was done for cVCR in Fig.1 and Fig.S1.

Also, I have a question regarding the subsampling approach. As described, subsampling was performed with "100 iterations of subsampling the vOTU count table at a range of different depths, spanning from 25 up to 16,000 counts at 50-count intervals."

Does the 25% mapping breadth threshold still apply in this subsampling process? Specifically, if removing just one read causes the mapping coverage to drop below 25%, would that vOTU still be considered? The method here is important, particularly to richness estimation, where rare viruses may be affected. I suggest that the authors provide more details on how subsampling was performed with the mapping threshold.

In addition, the usage of slope and direct subsampling points to compare diversity/richness across sequencing depths might be improved by using non-linear functions. For example, combining rarefaction and extrapolation using the iNEXT R-package could provide estimates of diversity/richness accounting for undetected viruses beyond observed data.

Given the big change in diversity results, I wonder whether another key metric, cVCR, and its associated results are also influenced by sequencing depth and normalization (subsampling). Since cVCR relies on the coverage of both viral and cellular genomes, low and varied sequencing depth could potentially bias cVCR values as well. So subsampling may potentially introduce some shifts in results, particularly in seasonal trends presented in Fig 1.

I believe the observed cVCR-based abundance seasonal pattern is biologically meaningful, but I would prefer authors to reinforce the conclusions by addressing these potential biases.

My suggestion is to justify the use of cVCR by testing whether cVCR values remain stable when reads are subsampled to the same sequencing depth.

2. As far as I understand, the parameter "-B 1000" used in the phylogenetic analysis, as written in the Methods section, refers to UFBoot. This should be explicitly stated in the Fig.5 legend, because different from traditional values, UFBoot values are only considered reliable if they are ≥ 95% (<http://www.iqtree.org/doc/Frequently-Asked-Questions>).

Additionally, it would be greatly appreciated if the authors could provide the full psb phylogenetic tree and alignment as supplementary files to allow readers to evaluate evolutionary relationships.

#3. The paper [<https://pubmed.ncbi.nlm.nih.gov/39448562/>] provides a background on polar viruses in a time-series context. The authors may consider referencing it to strengthen the introduction and discussion.

#4. I need some clarification for the newly added values in LN 145 because "On average, each sample harbored 10,215 ± 1,116 viral reads" (LN 142) and "The total number of reads per sample was 196,489 ± 18,358" (LN 138).

If the total reads per sample are ~196,489, how is the virus read proportion only 0.4% ± 0.5%? Additionally, when I checked Supplementary Table 1, the average number of viral reads per sample is ~800 (Column AN), which does not match the

reported $10,215 \pm 1,116$.

Could the authors clarify whether there is an inconsistency in these values?

#5. Fig. 2 needs clearer descriptions. For example, what does the error bar mean in the Fig.2B? And the of Fig2B seems to be “mean of richness values in subsampling” as I understand, but the legends said “b, Slope of richness rarefaction curves across sampling months.”, which is overlapped with the Fig S2.

Other legends could also be improved by providing more details. For example in Fig8A, what input data was used to calculate NMDS?

6. Viral taxa should be written in italics throughout the manuscript.

(Remarks on code availability)

Reviewer #2

(Remarks to the Author)

Review Nature Communications. NCOMMS-24-55589

Arctic Ocean virus communities: seasonality, bipolarity, and prokaryotic associations

I would like to sincerely thank the authors for their thorough and thoughtful responses to not only my comments but also those of the other reviewers. Their efforts to address the feedback provided have significantly improved the quality of the manuscript, and I appreciate the time and attention they dedicated to refining their work.

However, I would just like to highlight the following points:

L 488-490. Please clarify the following statement: “Notably, in the entire dataset of ASV to vOTU correlations, only one had a predicted host of Nitrosopumilaceae, which was correlated with an unclassified Gammaproteobacterium (no family-level prediction)”.

Does this mean that one virus has two putative host assignments, one being an archaeon (Nitrosopumilaceae) and the other a Gammaproteobacterium?

The discussion about the finding that many of Fram Strait viruses are also abundant in deep samples could include also the possibility that ocean currents in specific areas play an important role. The reason why these viruses are found at such depths remains puzzling.

Additionally, the term 'epipelagic' may be more appropriate than 'surface waters' when referring to depths of less than 200 m.

Regarding the new discussion paragraphs on how well these findings fit into host-virus evolutionary models, I completely agree with the authors that ecological models are often idealized in a chemostat-like setting. These dynamics primarily occur within seasons and should be evaluated through more spatially resolved temporal sampling.

However, I would like the authors to conduct a simple analysis: checking whether the exact same virus appears in consecutive summers, springs, or winters (not at 95%, if not at 100% ID). Are there any viral sequences with no genetic changes or SNPs observed across the temporal samples? The “Red Queen model” hypothesizes that 'fast' co-evolution occurs between hosts and viruses. If no genetic changes are detected, this hypothesis could be ruled out, at least within the analyzed period and possibly for a particular type of viruses. Instead, this would suggest the presence of a “viral seed bank” that persists in the environment between seasons but remains undetected due to the low abundance of its host at the change of season, influencing biodiversity in a way perhaps more aligned with the constant-diversity model. However, further analysis would be needed to confirm this hypothesis.

I have no other questions for this manuscript.

(Remarks on code availability)

The code is well-organized and easy to follow, which makes it a valuable resource for the community.

Reviewer #3

(Remarks to the Author)

The authors have addressed all comments to a satisfactory degree. I have no further concerns or suggestions and recommend this manuscript for publication.

(Remarks on code availability)

Version 2:

Reviewer comments:

Reviewer #1

(Remarks to the Author)

I appreciate the authors for the thoughtful replies and the updates. I have no further concerns.

(Remarks on code availability)

Reviewer #2

(Remarks to the Author)

Thank you for the detailed and thoughtful responses to all my comments and suggestions, as well as for the additional analyses and clarifications provided throughout the revised manuscript. I appreciate the effort made to address both methodological and conceptual aspects, including the clarification of host-virus associations, the discussion on deep-sea viral persistence, and the additional inStrain analysis on viral microdiversity. I have no further concerns and fully support the acceptance of this version of the manuscript. Just for the record, I would like to highlight my appreciation for the clarity of the revised discussion and the transparency of the supplementary data. Thank you again.

(Remarks on code availability)

REVIEWER COMMENTS

Responses in blue

Reviewer #1 (Remarks to the Author):

Calayag et al. present a well-conducted and comprehensive study of dsDNA viral ecology in the Fram Strait, Arctic, addressing a crucial gap in our understanding of virus dynamics in polar regions. The study demonstrates the seasonal structuring of viral communities over a four-year period, with distinct annual peaks in the virus-to-prokaryote ratio during the summer months. Additionally, the discovery that a significant proportion of Arctic viruses exhibit bipolar distributions further contributes to our knowledge of their biogeography. The use of convergent cross mapping network to identify virus-host interactions within ecological modules enhances our understanding of DNA viruses' ecological roles and interactions. Moreover, the analysis of amino acid signatures in viral proteins provides valuable insights into how viruses adapt to such extreme environments.

Overall, this study is valuable to environmental virologists, offering both important analysis results and data on the ecology of viruses in polar environments. The use of long-read metagenomics strengthens the findings by capturing larger genomic fragments. Additionally, the use of cVPR and CCM provides novel methods for understanding virus-host interactions. However, before this manuscript is considered for publication, I have some suggestions and comments listed below.

We thank the reviewer for their positive assessment regarding the value of our paper, and appreciate their comments as to the novelty of the methods and metrics we used. Below we address their suggestions and comments which we believe have improved the paper.

General Comments:

1. In Fig. 5b, many vOTUs do not encode structural or packaging-related genes, raising my concerns about the completeness of these cyanophage genomes. I wonder if this is an issue of labeling in this plot or if it reflects the nature of these genomes. It would be helpful if the authors could provide CheckV results along with basic statistics for the viral genomes to clarify genome completeness and quality.

We thank the reviewer for this constructive comment. Indeed, we only sporadically identified structural and other marker genes within the cyanophages. The goal was to highlight some genes to give the reader a sense of 'what is where', but not

systematically label as it would be too cluttered. Given that it is not really possible to do label exhaustively and the figure was already cluttered, we have removed this part, and replaced it with the much more quickly understandable CheckV (completion) results, as a barplot next to the heatmap to show the estimated completion of each respective vOTU.

We also now provide CheckV for all vOTUs as part of Supplementary Data 2.

Additionally, the bootstrap support in Fig. 5a for the basal topology appears low, with most nodes having bootstrap values < 80. To improve the reliability of this cyanophage phylogeny, I suggest the authors consider using alternative genes or incorporating cellular *psbA* genes into the tree.

We thank the reviewer for pointing this out. The *psbA* gene was utilized as it enabled establishing a phylogenetic relationship for all of the predicted cyanophage vOTUs, and also, as the reviewer points out, cellular organisms. We have now included cellular *psbA* and improved the bootstrap support for the tree, as well as a second form of support via Shimodaira-Hasegawa-like approximate likelihood ratio test. We now indicate where the cellular *psbA* are located in the tree in the main figure. The bootstrap support is still sometimes lower than ideal, but upon reviewing the literature, other versions of the *psbA* phylogeny often suffers from the same issue. For us, it was just important, mainly to somehow orient/cluster the most similar vOTUs together to give a sense of how closely related the cyanophage vOTUs are to one another. We now provide an update to the figure legend.

Figure legend:

Lines 1951–1957: **Fig. 5.** Cyanophages. **a**, *psbA* gene phylogeny of cyanophage vOTUs and reference genomes. Support values are reported as aLRT (approximate likelihood ratio test) / BS (maximum likelihood bootstrap). For visualization purposes, “*” indicates branches from this node are scaled 33% of their actual length for visualization purposes. **b**, Dynamics of correlated *Synechococcus* ASVs (top) and cyanophage vOTUs (bottom), along with completion estimates based on CheckV¹⁰⁸. ** indicates no completion prediction as the vOTU was classified as a provirus.

We update the methods accordingly.

Lines 926–957: Viral and prokaryotic reference *psbA* were extracted from the vConTACT2 viral reference database (ViralRefSeq-prokaryotes-v211⁵²) and GTDB prokaryotic reference database (version 214¹³²) by hmmscan¹³³ with PFAM Photo_RC. Eukaryotic reference *psbA* genes were extracted via Uniprot (gene_exact:psbA) AND

(taxonomy_id:2759, reviewed Swiss-Prot, n=157)¹³⁴. A preliminary tree was constructed from these sequences to differentiate between D1 (*psbA*) and D2 (*psbD*) photosystem II (PSII) reaction centre proteins, by alignment using MAFFT¹³⁵ followed by trimming with trimAl (-gt 0.2)¹³⁶, and FastTree¹³⁷ for phylogenetic reconstruction. *PsbD* were identified manually and excluded. The remaining sequences were considered *PsbA* and phylogenetic reconstruction performed a second time with the same alignment and trimming steps (but with -gt 0.8); at this stage, tree building was performed with IQ-TREE (settings: -B 1000, -alrt 1000, -m MFP)¹³⁸.

2. While Convergent Cross Mapping (CCM) appears to be a robust method for detecting causality in complex systems, I have some concerns about the application to lytic virus-host systems (high proportion in this study), where the short duration of interactions (with latent periods ranging from few hours to days) may complicate the inference of directional causality under a resolution of monthly sampling. This could potentially explain the limited overlap (4.3%; LN 221) between CCM associations and iPHoP predictions. I recommend that the authors provide further justification for their choice to use CCM in this context, potentially addressing these challenges in the methods section or adding a caveat in the discussion.

We acknowledge that the short duration of interactions in lytic virus-host systems, with latent periods ranging from a few hours to days, poses a challenge in inferring directional causality with monthly sampling resolution. We have chosen CCM because it offers a robust framework for detecting causality in complex ecosystems where traditional correlation methods may fall short. CCM allows us to infer potential causal relationships based on time-series data by analyzing the dynamics of both viruses and their prokaryotic hosts. The limited overlap (4.3%) between CCM associations and iPHoP predictions has been clarified in our revised manuscript. We hypothesize that this low overlap may be due to the temporal resolution of our data and the inherent biological variability in virus-host interactions. In the revised Discussion section, we elaborate on how different methodologies capture different aspects of viral dynamics, and we emphasize the need for further investigation to reconcile these findings.

Updated Discussion:

Lines 652–660: The use of CCM in studying virus-host interactions presents challenges due to their rapid temporal dynamics. Viral latent periods range from hours to days, complicating interpretations based on monthly sampling intervals. The low overlap (3.7%) between CCM associations and iPHoP predictions may reflect limitations of our sampling resolution. Integrating multiple methodologies is critical for understanding the complexities of microbial interactions. Combining CCM findings with naive network

analysis and iPHoP results provides a more comprehensive view of microbial community dynamics, with visual comparisons highlighting the biological significance of identified interactions.

Additionally, the use of certain parameters, such as a corrected Pearson correlation threshold (>0.7), may overlook biologically meaningful interactions, as viruses and their hosts do not always perfectly co-occur and gain a high correlation coefficient. A more detailed explanation of these parameter choices would be helpful to people who want to follow the study. Alternatively, the authors could present a naive network (without edge filtration) to compare against their CCM-based inferences, particularly regarding module clustering and mutual pairs identified by iPHoP.

Regarding the choice of parameters, particularly the corrected Pearson correlation threshold of >0.7 , we recognize that high correlation coefficients may not always represent biologically meaningful interactions. We have revised the Methods section to include a detailed rationale for our parameter choices, explaining how these thresholds were determined based on preliminary analyses and ecological relevance.

Methods, Lines 881–894: This threshold was determined based on previous findings on similar data^{58,124}, as well as preliminary analyses conducted on our dataset, which indicated that interactions exhibiting a correlation below this value often lacked biological relevance.

3. The amino acid composition analysis is presented as pooled data across all viral genomes, making it difficult for me to determine whether the observed amino acid signatures are driven by environmental adaptation or reflect community composition differences, especially given the strong viral endemism in polar regions. I suggest that authors consider analyzing amino acid traits in relation to phylogenetic distance, or constraining the analysis by same viral taxonomic groups, to better distinguish evolutionary lineage from environmental adaptation.

We appreciate the reviewer's point that the observed amino acid signatures could be driven by environmental adaptation or community composition differences.

These factors may be difficult to tease apart, but to improve our analysis, at the reviewer's latter suggestion, we have now re-performed separate analyses on the Caudoviricetes, Megaviricetes, and 'other'. Roughly 95% of the vOTUs are Caudoviricetes, but the patterns are the same across all groupings.

We update the text throughout to reflect the new taxonomy.

4. Authors have highlighted amino acid traits, but it would also be valuable to consider the role of other mechanisms, such as gene gain, in viral adaptation. Have authors examined whether there are gene orthologs specifically present or absent in polar regions? Or an additional discussion on how mechanisms may influence viral adaptations in polar environments could help readers understand virus adaptation to environments.

We appreciate the reviewer's point. We now include a table and a supplementary figure indicating the genes that are most enriched in high (greater than $> 60^\circ$ latitude N or S) and lower latitudes ($< 60^\circ$ degrees latitude).

This revealed some interesting patterns, which we highlight in the text:

Lines 588–595: Beyond the amino acid level, we found that 17.0% of viral protein annotations were significantly enriched in high latitudes (253 of 1,490), while another 7.1% were enriched in lower latitudes (106 of 1,490) (Supplementary Data 8). Notably, among a variety of annotations enriched at high latitudes, among the most significant were chaperone proteins and stress response such as Cold (CSD) and Heat Shock Proteins (HSP70), DnaJ (Supplementary Fig. 9), as well as genes common in cyanophages in the lower latitudes including a photosystem gene (Photo_RC) and Transaldolase/Fructose-6-phosphate aldolase (TAL_FC)⁷⁰.

Together, the annotation- and amino-acid-based enrichment analyses suggest that there are biochemical and biological traits distinguishing the polar viruses from the tropical and sub-tropical counterparts encoded in their amino acids.

Correspondingly, we add to the methods:

Lines 1035-1043: **Calculation of protein family enrichments by latitude**

For determination of proteins enriched in polar regions, we used the prodigal-predicted proteins for GOV2.0 and the Fram Strait virus dataset. These predicted proteins were annotated with pfam¹⁵² via hmmer¹³³ using the protein-specific cutoffs for significant (cut_ga). Then, individual protein families were summed within a given sample. Proteins that had prevalences of less than 10% were removed (n=1,490 remaining). Then, ANCOM-BC¹⁵³ was used to determine which annotations were enriched at latitudes greater than 60° N or S, or less than 60° N or S.

We also add a figure legend:

Lines 2060–2063: Supplementary Fig. 10: ANCOM-BC results showing protein families that are significantly enriched (top 10) at sampling locations of greater than 60 latitude ($p < 0.001$) or less than 60 latitude. The remaining list with p-values and significance scores as Supplementary Data 8.

Minor Comments:

LN135-139: The color palette in Fig. 2 makes it challenging to check the left panel. It would be helpful to use two distinct color sets between the two groups (Jan-May vs June-Sep).

We now use two distinct color sets for the two groups of months that have high and low diversity. The group of months with low diversity spans March and May, while the group with higher diversity spans August and February.

However, it should be noted that since the original submission we have re-run the alpha diversity analysis because we realized that it is better to rarefy the data and re-calculate the statistics. This analysis revealed slightly different results for the alpha diversity analysis, resulting in higher richness of vOTUs in late summer-autumn (Aug - Nov), and lower diversity in March and May. However, large variations in richness values observed within months suggests variable viral dynamics.

We update the manuscript accordingly in the results (as long as changes to the methods).

Lines 194–207: As viral diversity was not saturated at any sequencing depth (Fig. 2a), we compared the diversity of vOTUs across months after subsampling to a normalized depth. The diversity of viruses increased following the spring phytoplankton bloom, reaching a maximum between late summer-autumn (Aug–Nov) (Fig. 2b), with lowest richness values in March and May. Notably, prokaryotic community richness was also lowest in May, though the timing of highest prokaryotic richness was different than for viruses, with the highest prokaryotic richness observed in winter (vs. late summer-autumn for viruses)⁴¹ (Supplementary Data 1). For the virus richness measurements, although these patterns were derived from subsampled counts, we observed that the fraction of viral diversity captured in winter was much lower than in late summer, as indicated by the slopes of rarefaction curves (Supplementary Fig. 2). These differences, in part, reflect variations in sequencing depth across samples and suggest that further sequencing efforts are required to unravel the viral diversity that persists through late polar night.

LN142: The lower diversity and slower accumulation observed in summer could be a result of higher overall biomass during that season. Few dominant bacteria and viruses may account for the majority of reads at the current sequencing depth (10Gb), so the detection of rare viruses has become relatively difficult. Please consider adding this as a potential limitation in the text.

After rarefaction, summer now has higher diversity, along with winter, and it is spring that has the lowest diversity in general. As the reviewer points out, this could be due to the dominance of a few abundant viruses, and is challenged by the relatively low sequencing depth, and mention this as a caveat to these aspects.

Lines 194–207: We next investigated how the overall diversity of viruses changes across seasons. As viral diversity was not saturated at any sequencing depth (Fig. 2a), we compared the diversity of vOTUs across months after subsampling to a normalized depth. The diversity of viruses increased following the spring phytoplankton bloom, reaching a maximum between late summer-autumn (Aug–Nov) (Fig. 2b), with lowest richness values in March and May. Notably, prokaryotic community richness was also lowest in May, though the timing of highest prokaryotic richness was different than for viruses, with the highest prokaryotic richness observed in winter (vs. late summer-autumn for viruses)⁴¹ (Supplementary Data 1). For the virus richness measurements, although these patterns were derived from subsampled counts, we observed that the fraction of viral diversity captured in winter was much lower than in late summer, as indicated by the slopes of rarefaction curves (Supplementary Fig. 2). These differences, in part, reflect variations in sequencing depth across samples and suggest that further sequencing efforts are required to unravel the viral diversity that persists through late polar night.

Fig. 3b: Using different shapes for the dots to represent different years could make it easier to understand temporal trends across four years.

Figure 3b has been revised to distinguish years based on different point shapes.

LN168: I'm uncertain whether the phrase "shaped by their hosts" is appropriate, as it's difficult to attribute the observed viral patterns to direct host influence, given the potential roles of other prokaryotes.

Lines 275–279: These results suggest that viral communities are primarily correlated with prokaryotic community composition which is in turn driven by environmental conditions plus biological interactions – leading to a complex network of interdependencies and physicochemical linkages, similar to dynamics in temperate environments^{7,49}.

Additionally, I am a bit confused about which pair in Supplementary Data 1 represents the structure of viruses in the main text. Authors could add the values for each variable in the main text.

The reason we performed multiple sets of Mantel tests was because not all viruses are predicted to be viruses of prokaryotes, yet we correlated the entire virus community with the prokaryotic community because it is the most comprehensive analysis to perform. But recognizing some of the viruses may be viruses of eukaryotes, we wanted to see also what the correlations were if we excluded likely viruses of eukaryotes. Thus, we also correlated only the Caudoviricetes with the bacterial community, and the other environmental parameters for good measure. This tests showed the differences were negligible, thus we focus our main analysis on the entire virus community (most comprehensive), which is column 1 (All vOTUs), and thus we didn't want to confuse the main text by including so many values, that don't add much to the main take-away message (i.e., suitable for the supplement).

We now provide a more thorough legend for Supplementary Data 1 (now Supplementary Data 3) to explain this result, along with environmental parameters. We also embolden and mark with ** the columns used in the main text.

Additionally, we have added a new figure to highlight one the full virus community composition to bacterial community, along with correlating it with various other parameters as a network. Here, we also cite Supplementary Data 3 as supporting information.

Lines 268–271: At the whole community level, Mantel tests demonstrated the strongest correlation to prokaryotic community composition (Mantel $\rho = 0.632$, $p = 0.001$) followed by oxygen (Mantel $\rho = 0.371$, $p = 0.001$) and mixed layer depth (MLD) (Mantel $\rho = 0.259$, $p = 0.001$) (Supplementary Data 3, Supplementary Fig. 4).

LN179: Please add the number of 'following' and 'caused by'.

We clarify:

Lines 286–325: The CCM network comprised 5,136 vOTUs and 850 prokaryotic ASVs (Supplementary Fig. 5a), with directional associations (correlations >0.7) where vOTUs dynamics are 'following', or are 'caused' by, prokaryotic ASV dynamics (akin to Lotka-Volterra dynamics⁵⁰) totalling 15,930.

LN192-193 & Fig. 4d: The ratio of vOTUs/ASVs in M5 is significantly lower than in other modules, and the value is somehow counterintuitive. Could this be due to that most winter viral nodes being discarded during network preprocessing? A bit further discussion on potential reasons behind this observation would be helpful.

The reviewer makes an astute observation that vOTU to prokaryotic ASVs is significantly lower in M5. This is the module which contains the most prokaryotes. Yet, this period of low number of vOTU also corresponds to the period at which overall cVCR (i.e., the coverage-based virus to cell ratio) is lowest. Thus, yes, it could be due to lower sensitivity to viruses during this period. However, it would be difficult to quantify if viruses that otherwise might appear in M5 were removed from the network analysis due to preprocessing, as we can't predict which viruses would be in which module without running new networks with different settings etc, which may be less robust.

We now provide a bit further discussion on potential reasons for this observation in the discussion:

Lines 560-566: It should be considered that factors such as diversity and abundance can impact the total number of viruses detected in modules, and co-occurrence patterns overall. M5, for example, which has the lowest vOTU to prokaryotic ASV ratio, corresponds to the period during which vOTUs were the least relatively abundant. Additionally, during this period, vOTU richness was relatively high. The combination of these two factors challenges vOTU detection, and could result in vOTUs being excluded during pre-processing or missed altogether.

LN265: Are the 58 vOTUs mentioned evolutionarily related (similar taxa)?

Lines 486-490: In total, 53 of the 58 vOTUs were classified as Caudoviricetes (the other five were unclassified), and came from 14 different novel orders (Supplementary Data 6). Notably, in the entire dataset of ASV to vOTU correlations, only one had a predicted host of Nitrosopumilaceae, which was correlated with an unclassified Gammaproteobacterium (no family-level prediction).

Fig. 4C: Please ensure the p-values have been corrected for multiple testing.

The p-values have been adjusted for multiple testing using the Bonferroni correction method.

Supplementary Data: It is not easy to understand the headers in supplementary data 1 and 2. Please consider using one spreadsheet to include descriptions of the column headers for all tables.

We have added descriptions of the column headers below relevant Supplementary Data tables to improve clarity.

Reviewer #2 (Remarks to the Author):

Review Nature Communications. NCOMMS-24-55589

Arctic Ocean virus communities: seasonality, bipolarity, and prokaryotic Interactions
This study investigates the seasonality, abundance, diversity, associations with prokaryotic hosts, and environmental factors of Arctic viruses in the Fram Strait over a four-years period. The authors used 47 long-read metagenomes obtained from the cellular size-fraction, which were automatically collected. It was found that viruses exhibited annual peaks, and that the composition of the viral community was strongly correlated with prokaryotic community composition. Additionally, it was analyzed presence of the polar virus in metagenome from different latitudes, concluding that virus abundance is higher in high-latitude regions of both poles, highlighting adaptations for cold environments.

First, I would like to positively emphasize the rigorous mathematical and statistical methods employed by the authors in this work. While seasonal cyclic variability in viral communities has been observed in prior studies, the extended sampling period here provides a more robust dataset than previous research. The constructed metagenomes alone represent a valuable resource database. However, some sections lack sufficient biological interpretation and related discussion. Further clarification from the authors on the following specific points is recommended to enhance the overall quality of the manuscript.

We thank the reviewer for their positive evaluation of our statistical methods and appreciation of the novelty of the analysis. We appreciate the opportunity to clarify the points below.

In general, more quantitative data should be included in the main text and as Supplementary Data.

We now have incorporated additional quantitative data in the main text (Lines 141–142, Lines 153, Lines 179–180, Lines 331–333, and Lines 340–341), and have expanded

Supplementary Data 1 to include comprehensive metadata. This metadata now covers environmental parameters, project identifiers (bioproject and accession codes), and taxonomy-based read counts and proportions across the domains Archaea, Bacteria, and Eukarya.

Table of Supplementary Data 4, in particular, could be expanded to include specific information such as the number of viral sequences in each collection, the depth (or range) of the water column from which each sample was obtained, the maximum length of the viral raw reads obtained (L239), the estimate of the number of prokaryotic genomes in each metagenome, and, if possible, parameters indicating viral (L142) and prokaryotic diversity.

We have expanded Supplementary Data 4 (now Supplementary Data 1) to include the depth at which each water sample was collected, the Bioprojects, Accession numbers, as well as per-sample data on the number of viral contigs, the length of the longest viral contig, the number of ASVs, and the read counts and proportions of archaea, eukaryotes, prokaryotes, viruses, and organelles as predicted by Tiara v. 1.0.3.

Additionally, a global overview of prokaryotic taxonomy retrieved by ASVs and their periodicity is needed in the Supplementary Data to improve the interpretation of the seasonality observed in the viruses (L142, L80).

We now add the rarified statistics for the prokaryotic diversity referenced in L142 to our Supplementary Data 1. This Prokaryotic diversity from the same time-frame is reported in (Priest et al. 2024). We now cite Priest et al. 2024, as well as additional references and point directly to the results in these papers, and point out that those results also show that richness is lowest during the spring.

Lines 199–202: Notably, prokaryotic community richness was also lowest in May, though the timing of highest prokaryotic richness was different than for viruses, with the highest prokaryotic richness observed in winter (vs. late summer-autumn for viruses)⁴¹(Supplementary Data 1).

Details regarding the sizes and number or replicas of the 16S rRNA libraries should also be provided as Supplementary Data).

We now provide the number of raw sequences and number of replicates (in all cases $n=1$), and accession numbers for the 16S rRNA libraries that accompany the metagenomic samples as part of Supplementary Data 1.

Furthermore, data on the ASV-vOTUs involved in the six modules, as well as those identified by iPHoP (L181 & L214), must also be included.

We have now provided more information about the ASV-vOTUs involved in the six modules, comparing and contrasting the results more explicitly between host predictions (iPHoP) and correlations between vOTUs and ASVs (CCM). We add two additional figures, one additional table showing the cases in which they agree, as well as more in-line descriptions of the results:

Lines 363–413: Among bacterial families with a high number of predicted vOTUs, Flavobacteriaceae and Pelagibacteraceae were the most frequently predicted. In M1, Flavobacteriaceae and Akkermansiaceae (Verrucomicrobia) dominated, while Pelagibacteraceae was less common. Conversely, in M5, Pelagibacteraceae was the most predominant, with fewer Flavobacteriaceae and Akkermansiaceae. Meanwhile, vOTUs with Cyanobiaceae as their predicted host were more common in M2–M4 (Supplementary Fig. 5d). Among the vOTUs correlated with an ASV in the network, the majority of host-virus pairwise correlations were to diverse prokaryotes present in M5, with Pelagibacteraceae (20.3%) and Nitrosopumilaceae (6.0%) being the most common (Supplementary Fig. 5e). M1 had the second most bacteria-to-virus connections with 37.5% of these being to Flavobacteriaceae. Among the families with most pairwise correlations (i.e., over 250), the overall pattern of host prediction of the viruses and module membership was similar, with vOTUs with predicted hosts of Flavobacteriaceae and Pelagibacteraceae being the dominant host predictions for M1 and M5, respectively (Supplementary Fig. 5f). However for individual vOTUs and ASVs, only 3.7% of the predictions overlapped at the family level between CCM network correlations and iPHoP (i.e., 122 out of 3,280 correlations with associated host predictions). Among these, again, Flavobacteriaceae and Pelagibacteraceae made up 46.7% and 6.6%. For Flavobacteriaceae, these vOTUs were diverse, coming from 14 different novel orders (via VConTACT⁵², see Methods), while for Pelagibacteraceae they all came from a single novel order (Supplementary Data 4).

To better grasp the magnitudes involved (as the authors provide statistical data but almost no “raw numbers”), it would be helpful to include precise information in the main text. For example, it would be useful to know how many viral sequences were identified among the nearly 200 000 sequences obtained per metagenome (L96); how many viral sequences represent the 5662 vOTUs identified by the authors; which vOTU was the most abundant and how it varied along time and how many vOTUS are involved in each module (L173-195). Additionally, the 3748 ASVs mentioned in L391 should be cited earlier, at the beginning of the Results sections.

We have incorporated the following information, in response to the aforementioned questions, along with other specific details where relevant:

Lines 141–142: "...10,215 ± 1,116 viral reads and 180 ± 29 viral contigs."

Lines 153: "...which were derived from 7,775 viral contigs that were greater than 10 Kb in length."

Lines 179–180: "...the most abundant vOTU, a member of the Caudoviricetes, was also the most persistent, occurring in 45 of the 47 sampled time points."

Lines 331–333: "...M1 containing 1,448 and M2 containing 1,334 vOTUs, respectively."

Lines 340–341: "...M5 also contained the fewest vOTUs, with only 132 identified."

We now also mention the 3,748 ASVs at the beginning of the **Virus-host and environmental relationships** subsection of the Results.

Lines 265–266: "...comprising 3,748 prokaryotic amplicon sequence variants (ASV)."

Specific questions:

1.-In my opinion, the title of the article is inadequate. First, not all viruses from the community were considered, as RNA viruses were excluded from the analysis (L422), and the authors focused primarily on dsDNA viruses. Importantly, the authors examined viruses found exclusively in the cellular fraction; i.e., those actively infecting cells at the sampling points or integrated into cellular genomes as lysogens. Additionally, there is limited information on prokaryotic "interactions" per se, as host identification relies exclusively on computational predictions. I believe a title emphasizing the seasonality and the dynamics of the viral community would better reflect the study's scope and attract a broader audience. Moreover, only viruses present in the cellular fraction are considered in the analysis, rather than the complete community- an important detail that should be more clearly stated throughout the manuscript.

We thank the reviewer for feedback on the title. We are concerned that dsDNA may be a less accessible qualification to the title. We mention dsDNA in the third sentence of the abstract. We change the title to 'prokaryotic associations' rather than prokaryotic interactions, since perhaps associations implies less depth of analysis. We update the title to:

Arctic Ocean virus communities: seasonality, bipolarity, and prokaryotic associations

Lines 634–637: A further consideration is that our study focused on dsDNA viruses, thus not investigating the RNA virome^{81–85}. Investigating RNA viruses, their seasonality and impacts on the prokaryotic community is an important future direction.

2.-The introduction provides quotes related to the 'Arctic Atlantification', but this information is not directly relevant to the goals or findings of this manuscript. In addition to the hypothesis proposed by the authors, L79, which is not entirely novel (reference 22, 25), I would encourage the authors to present hypotheses in the introduction regarding how polar viral communities would fit in the ecological models already described for virus-host populations., e.g., 'Piggyback-the-Winner' theory, Red-Queen dynamics, Constant Diversity Model, etc. (F.R. Valera et al. 2009, B. Knowles et al. 2016, F. H. Coutinho et al. 2017, H. Alrasheed et al. 2019, J.C. Ignacio-Espinoza et al. 2020). This would help frame the discussion to better align with the Lotka-Volterra dynamics observed.

We thank the reviewer for helping with the framing.

We clarify in the section about cyanophages that cyanobacteria are expected to increase in the Arctic due to Atlantification, so that these observations are related to this:

Lines 84-116: Furthermore, continuous sampling, over the long-term, can discern the impact of 'Arctic Atlantification'³⁴⁻³⁶: the northward expansion of subarctic habitats through the Fram Strait — the major connection between Atlantic and Arctic Oceans³⁷. Here, polar water outflowing the central Arctic Ocean via the East Greenland Current (EGC) meets the West Spitsbergen Current (WSC), transporting warmer Atlantic water into the Arctic Ocean³⁸⁻⁴⁰. In the WSC, prokaryotic communities exhibit pronounced seasonality, underpinned by changes in photosynthetically active radiation and mixed layer depth⁴¹.

Lines 423-425: From the predicted interactions between vOTUs and ASVs, we further explored the diversity and dynamics of putative cyanophages – as Cyanobacteria are of particular interest with respect to changing conditions in Arctic ecosystems^{5,53,54}.

Lines 473-476: Our results reveal that both cyanobacteria and their viruses are present in the Arctic, highlighting a need to further understand their interactions for the future Arctic Ocean, as they are expected to increase due to Atlantification.

Furthermore, we address the concerns raised about the hypothesis and mention the question of how polar viruses would fit ecological models described for virus-host populations described elsewhere:

Lines 118–127: Given this, and the tight coupling of viruses and their hosts^{9–11}, we aimed to examine the degree that viral populations are seasonally structured in the Arctic. The strong seasonal gradients in the Arctic provide an opportunity to observe host-virus dynamics and how they relate to prevailing conceptual models such as the Piggyback-the-winner^{42–44}, Constant-Diversity^{45,46}, and Red-Queen^{28,47} hypotheses. We examined the diversity and seasonality of dsDNA virus communities in the Arctic Fram Strait over four complete annual cycles at roughly monthly resolution. This unprecedented time-series of virus communities in the Arctic demonstrates considerable seasonality in viral diversity, their association with environmental conditions and potential microbial hosts, and their distribution across the global ocean.

We also frame the results within the context of these concepts:

Lines 698–715: Our results indicated a low percentage of predicted prophages without a seasonal enrichment, even in the cellular size fraction studied. This suggests that the “Piggyback-the-winner” scenario is neither very prevalent, nor seasonally variable, though again this may be currently limited by computational predictions, and other biases like lytic viruses occurring within cells in more than one copy, and thus we may underestimate prophage prevalence. We found that most viruses are part of seasonally variable modules, and that often there is no similarity between opposing seasons (e.g., six months apart), at least at the depth sequenced at in the investigated cellular size fraction. Hence rather than having a ‘Constant-Diversity’^{45,46}, Arctic virus communities are undergoing substantial seasonal change. Likewise, the prokaryotic community is also highly seasonal, thus the viruses are generally following and strongly correlated with host abundances. Thus, unlike how viral ecological models are sometimes idealized in a ‘chemostat-like’ setting¹⁰⁰, viral communities’ strong seasonality impact how such models can be of use in the Arctic Ocean (such as in the Red-Queen hypothesis^{28,47}). Arctic virus communities, with their strong seasonality where the many microorganisms are not present year-round (or at least often not detected), may be impacted by how these dynamics play out elsewhere, or it’s likely these dynamics occur primarily within seasons. In each case, these dynamics could be further evaluated via more spatially-resolved temporal sampling.

3. There are two points that need to be clarified regarding the methodology used:

(i) Importantly, the authors claim that the lytic lifestyle was predominant over lysogeny.

Although I find value in the analysis conducted by the authors, I have a primary concern regarding the number of lysogens detected. Beyond bioinformatic predictions, the authors lack direct evidence to fully support this conclusion. Many lysogens do not have

recognizable integrases, which is a general challenge that needs to be addressed (out in the context of this work). In my view, viruses replicating inside the cells (i.e., lytic viruses) would outnumber lysogens in term of copies within the cellular metagenome, making them easier to assemble with the long-reads used, compared to lysogens. Therefore, a strong bias could obscure the results.

We appreciate the points about lysogens potentially not being recognized due to recognition limitations and about copy numbers of lytic infections vs. lysogens.

State-of-the-art lysogen/prophage predictors (VIBRANT, CheckV, and geNomad) use a variety of techniques to identify lysogens, most typically looking for areas of contigs that are enriched in viral genes surrounded by proteins that appear to be of cellular origin. Originally, we used VIBRANT to predict lysogens, but upon critically evaluating the point by this reviewer, we realized our workflow for predicting lysogens was flawed because we applied VIBRANT to contigs from which CheckV had already been used to ‘trim off’ host regions of prophages. We have now rectified this issue and replotted the data for Fig. 1c, and Fig. 5c, though the main result, that lysogens/prophages are rare, remains the same.

We have added the following sentences to aid in the interpretation and describe the limitations.

Lines 184–192: Overall, lytic lifestyle was predicted to be predominant, with an average of $94.0 \pm 0.4\%$ of viruses, a pattern that was invariable across seasons (Fig. 1c). The Southern Ocean featured higher rates of lysogeny during periods of low production²⁵. As for the latter point, it is important to point out that some lysogens may not be recognized due to missing integrases or other factors, and, vOTUs may not represent the full prophage, which could then be misinterpreted as lytic rather than integrated. Additionally, lytic viruses may outnumber lysogens in terms of copies within the cellular metagenome, making them easier to detect and assemble, which could bias the recovery or obscure lysogens.

(ii) Cut-off used to define the abundances of vOTUs. The authors use only 25% of the viral genome length, whereas typical values range from 50 to 70% at a minimum.

We appreciate this point by the reviewer, and we had discussed this amongst co-authors previously. We evaluated different cut-offs and decided for 25% as a trade-off between confident genome coverage and sensitivity, and based our decision as follows. First, the reads we are using are long-reads with an average length of $5,435 \pm 405$. Thus, this can be considered when defining the minimum coverage, especially as it

relates to the more common short-read metagenomes. For short reads, unlike long-reads, the 150 bp reads could map to multiple conserved regions across many genes of a contig, which may be more problematic and predictable in terms considering that most genes have highly conserved motifs and less conserved motifs (circumvented by long-reads). Additionally, considering that the shortest contig that we decided to use was 10kb (which was a conservative cut-off, itself), on average, the minimum coverage would be 50%, though some may be less than that.

Nevertheless, we considered how different cut-offs can influence our results, especially the main result of seasonality within the dsDNA virus community. **We found that the seasonality is similar across a variety of cut-offs from 0, 0.25, 0.5, and 0.75.** The difference is that the higher the coverage required, the less resolution for the number of viruses that can be observed in any given sample. Given the fact that we had long-reads and that the main result of seasonality still strongly held at various cutoffs, we felt comfortable with 0.25 which would enable more resolution of viruses across months, even where they may be more rare.

We have updated the manuscript with a supplementary figure showing how the various cut-offs do not impact the result of seasonality.

Lines 257–259: Notably, the seasonality observed in virus community composition was consistent regardless of minimum coverage value utilized for detected viral presence, though similarity was less at higher thresholds (Supplementary Fig. 3).

4. The authors focus on Cyanobacteria and the dynamics of putative cyanophages (L232-257) due to their relevance to changing conditions in Arctic ecosystems (but it is unknown in which module are founded), along with another nice example of the Nitrososphaerales. In my opinion, there is a lack of detailed information regarding the largest group, which is represented in several modules and with multiple identified virus-host pairs, the Pelagibacteraceae. Providing information on how the Pelagibacteraceae ASVs are distributed across the different modules, along with the assignment vOTUs, would better complement this section of the manuscript.

We have now gone into more detail on the Pelagibacteraceae dynamics, as well as other dominant hosts, the Flavobacteriaceae; two groups which dominated the CCM networks. We provide a new chord diagram that indicates that most of the ASV-to-vOTU correlations occur to prokaryotes in M1 and M5; these patterns are similar to the host predictions of the vOTUs in these modules. We also provide information specifically about the vOTUs that are both correlated with and have host predictions for Pelagibacteraceae (Supplementary Data 4).

Lines 362–418: In total, 42.5.2% of vOTUs received a host prediction by iPHoP. Among bacterial families with a high number of predicted vOTUs, Flavobacteriaceae and Pelagibacteraceae were the most frequently predicted. In M1, Flavobacteriaceae and Akkermansiaceae (Verrucomicrobia) dominated, while Pelagibacteraceae was less common. Conversely, in M5, Pelagibacteraceae was the most predominant, with fewer Flavobacteriaceae and Akkermansiaceae. Meanwhile, vOTUs with Cyanobiaceae as their predicted host were more common in M2–M4 (Supplementary Fig. 5d). Among the vOTUs correlated with an ASV in the network, the majority of host-virus pairwise correlations were to diverse prokaryotes present in M5, with Pelagibacteraceae (20.3%) and Nitrosopumilaceae (6.0%) being the most common (Supplementary Fig. 5e). M1 had the second most bacteria-to-virus connections with 37.5% of these being to Flavobacteriaceae. Among the families with most pairwise correlations (i.e., over 250), the overall pattern of host prediction of the viruses and module membership was similar, with vOTUs with predicted hosts of Flavobacteriaceae and Pelagibacteraceae being the dominant host predictions for M1 and M5, respectively (Supplementary Fig. 5f). However for individual vOTUs and ASVs, only 3.7% of the predictions overlapped at the family level between CCM network correlations and iPHoP (i.e., 122 out of 3,280 correlations with associated host predictions). Among these, again, Flavobacteriaceae and Pelagibacteraceae made up 46.7% and 6.6%. For Flavobacteriaceae, these vOTUs were diverse, coming from 14 different novel orders (via VConTACT⁵², see Methods), while for Pelagibacteraceae they all came from a single novel order (Supplementary Data 4). However, overall, the low number of matching CCM correlations and host predictions at the vOTU-ASV level demonstrates the challenge to discern host-virus relationships in diverse and complex ecosystems where a variety of technical and biological challenges may complicate such analyses (see Discussion). Nonetheless, the results provide valuable indications for particular lineages at an overall module level.

Other details:

Just as a suggestion, did the authors consider whether Supplementary Figure 3c could appear as a main figure (perhaps in place of Figure 4a)?

We have now moved it to the main text and moved the original Figure 4a to the supplement.

L63. Please, clarify. Sandaa et al 2018, (reference 22) found that (cited textually) “Viral diversity and virus-to-prokaryote ratios (VPRs) dropped sharply at the commencement of the spring bloom but increased across the season, ultimately achieving the highest levels during the winter season. These findings suggest that viral lysis may be an

important process during the polar winter, when productivity is low". Therefore, this contradicts what is stated in L59, 60.

We reviewed these papers again, and have revised the manuscript accordingly:

Lines 69–71: One year-round study showed strong seasonal variation among virus communities, with sharp decreases in virus-to-prokaryote ratios at the onset of the spring bloom and highest ratios in winter²² Others have reported increases during spring-summer, and highest abundances of viruses in winter^{23,24}.

We also add a line to the discussion:

Lines 611–615: Our observation of strong virus seasonality in the cellular size fraction contrasts with some of those of free viruses via fluorescence microscopy. This seeming contradiction may indicate a difference in the dynamics of infection and host-association, or loss factors of free-living viruses^{71,72}, for example photo-degradation of free viral particles^{73,74}.

L93-94. Did the authors retrieve eukaryotic DNA in their sequencing? Since no pre-filtering was applied, there is a high possibility of this occurring, yet it is not mentioned in the manuscript. If present, could this affect the cVPR calculations?

We have now evaluated the amount of eukaryotic DNA in the samples, and find that eukaryotic reads made up $1.5\% \pm 0.53\%$ of reads in the metagenomic samples, vs $72.2\% \pm 15.3\%$ of prokaryotes, indicating that eukaryotic sequences made up a small minority of the samples, even without pre-filtration.

We update the Results:

Lines 142–145: Notably, despite not pre-filtering to remove eukaryotes, the large majority of reads in the dataset were prokaryotic ($72.2\% \pm 15.3\%$), with eukaryotes and viruses making up smaller proportions ($1.5\% \pm 0.53\%$, and $0.4\% \pm 0.5\%$, respectively).

Discussion:

Lines 609–611: This increase was consistent regardless of the normalization used, namely either coverage-based virus to cell ratio (our main metric), or also coverage per gigabase pair (which would also account for eukaryotic DNA).

And Methods:

Lines 849–852: To gain an insight into the composition of the metagenome samples overall in terms of prokaryotic and eukaryotic reads, we employed Tiara v.1.0.3¹²¹, a machine-learning tool, to assign domain-level classifications to the raw reads. This revealed that the large majority of reads across the metagenomes, >95%, were of prokaryotic origin.

The reviewer has made a good point in asking how the cVPR can affect the cVPR values. In fact, upon further consideration, we realized that our cVPR metric actually does include eukaryotes within the calculation, as it calculates the average coverage of 16 universal ribosomal proteins found across all three domains.

We have edited the methods to clarify this point throughout the manuscript, and renamed the metric cVCR (coverage-based Virus to Cell Ratio), from cVPR (coverage-based Virus to Prokaryote Ratio). However, again, the eukaryotes are making up a small amount of the ratio itself since they only make up ~1.4% of total reads.

We update the methods:

Lines 842–844: Second, we divided the mean coverage by the estimated number of cellular genomes in each metagenome, as predicted based on the mean coverage across 16 universal single-copy ribosomal proteins^{119,120}.

L98-99. The program CheckV is mainly used to confirm the nature of the viral sequences, not to make predictions per se. Therefore, remove from this form the line, and then the two approaches mentioned in L99 that refers the authors, will be correct.

We appreciate this point by the reviewer and agree. We were following a published VirSorter2-based standard-operating-procedure for virus prediction that includes CheckV. We now specify this more fully and cite the SOP that we based our analysis on: <https://www.protocols.io/view/viral-sequence-identification-sop-with-virsorter2-5qpvoyqebg4o/v3>

To avoid confusion, we remove the tools from the results and cite the methods where we can describe this more fully.

Lines 139–141: Viral sequences were predicted on both assembled contigs and raw reads (see Virus Prediction, Methods) and the overall number of predicted viral sequences was concordant between the two approaches (Supplementary Fig. 1).

L107. “95% sequence clusters”. Please modify to indicate that this refers to 95% ID.

We edited, Line 152: “95% nucleotide identity clusters”

L109. Please note that it is recommended to follow the updated nomenclature for phages. The bacteriophage families Myoviridae, Podoviridae, and Siphoviridae have been reclassified under a new taxonomic system (International Committee on Taxonomy of Viruses (ICTV), 2021). These are now part of the order Caudoviricetes, which encompasses tailed bacteriophages. Myoviridae members are often reclassified in families like Myoviricetes, while Podoviridae members generally fall under Autographiviridae, and Siphoviridae have been split across families like Drexlerviridae and Demereciviridae, among others.

Indeed the ICTV is ratified/accepted, but the taxonomy and tools and strategies for employing it are still being developed, especially for uncultivated viruses. This includes ongoing work within the Caudoviricetes, the Class of 98% of the viruses in our manuscript. There is only one tool, geNomad, that provides high-quality classifications based on the ICTV at this point, but this tool did not more finely resolve most of our Caudoviricetes sequences beyond the level of Caudoviricetes. vConTACT3 could provide more information regarding protein network structure (and theoretically, taxonomy), but the results were not clearly easy to incorporate, with 91.3% coming from novel orders and only 0.4% receiving a taxonomy at the order level (Crassvirales). Nevertheless, we provide the vConTACT3 taxonomies as part of Supplementary Data 2, and cite them in cases where we talk about specific vOTUs.

We now apply geNomad to update our taxonomy to the ICTV taxonomy, which gives overall very similar results to our initial submission, i.e., if one were to ‘add-up’ the different out-dated morphological families (Myo-,Podo-,Sipho-) to ‘Caudoviricetes’. If we had focused on differences within Caudoviricetes (at a finer taxonomy), then this update would potentially have caused more issues, but as we did not, it's not such a big problem (as also mentioned by the reviewer).

We update the methods, and recalculate values for class level variability we discussed, as well as regenerate figures to include the new taxonomy.

Overall, the main conclusions and patterns remain unchanged.

L111. Phycodnaviridae continues to be cited in in L122, L199 and Figure 3, so please, update this sentence accordingly.

We have removed the part of the sentence about not focusing on Phycodnaviridae (now included in 'Megaviricetes' in the paper).

L166. Please define 'PAR' as it is the first time it appears in the text. Also, what parameter exactly is the "polar water fraction" in Figure 3b? Additionally, "iceDist" and "icePast" are not mentioned in the manuscript (also in Figure 4).

We have defined "PAR" (Photosynthetically Active Radiation) (Lines 273–274) as suggested.

Polar water fraction is the proportion of polar water in the sample and was derived from salinity and temperature. We now describe this in the Methods (Lines 763–765).

Additionally, we have removed iceDist and icePast from the manuscript, as these parameters are less relevant in the context of the West Spitsbergen Current (WSC).

L287. Is this in disagreement with the statement in L280? First, authors must clarify whether the vOTUs detected in surface waters are different from those detected in deep waters. If Fram Strait viruses are also abundant in deep ocean samples, does this imply that similar hosts are more abundant at these depths? Additionally, which depth range is being referred to? This information needs to be either depicted in the figure or stated in the Supplementary Data 3. It is generally understood that deep and surface prokaryotic communities differ... then, could mixed waters be influencing this observation (although the authors specifically mention that these "mixed" samples were not considered in the analysis)? The authors should provide a more detailed discussion on this point (L291-298).

We have revised this section for clarity. We didn't mean to imply that the Fram Strait viruses were more abundant in the deep samples, only that the proportion of samples in which they were detected was higher in deep samples. In terms of abundance, Fram Strait viruses are more abundant in the surface waters, but often completely absent from surface ocean samples (for example in the low latitudes < 30 N and S).

We now also clarify that the communities of viruses present in surface vs. deep do differ significantly in terms of 'who is where' which we now point out via an adonis test which can detect differences in community structure between two groups.

We clarify these points as follows:

Lines 537–548: Fram Strait viruses were also common in the deep ocean (here, operationally defined as sampling depth of >200m). In particular, they were detected in 87% (338 of 390) of deep global samples. Like the surface waters, deep-water viruses were more prevalent in northern and southern higher latitudes (Fig. 7b), though in deep samples the viruses were more commonly detected. Overall, however, when detected, their relative abundances were less in deep than in shallow samples. Notably, there was significant variation in community structure of the detected Fram Strait viruses between surface and deep samples (adonis2, Sum of Squares = 38.68, $R^2 = 0.11448$, $p < 0.001$), which aligns with established differences in microbial communities in the deep vs. surface ocean, aligning with past results of virus biogeography¹⁴. Future work could consider the differences in the total virus communities between surface and deep, as the current analysis utilizes only mapping to the Fram Strait viruses.

We also add to the methods to describe how the adonis2 results were calculated:

Lines 980–985: For evaluating statistical differences in community similarity between surface (<200m) and deep (>200m) samples, we first removed samples that had less than 0.0001 cVGB (n=769 samples remaining, including 435 and 334, shallow and deep samples, respectively) and vOTUs that had less than 0.0001 summed cVGB across all samples (n=4,693 vOTUs remaining). Then, we ran adonis2^{122,148} with 999 permutations based on the surface and deep grouping.

L310-L317. Please, provide more details on the changes observed in the amino acid composition of the viruses from Fram and GOV2.0 datasets versus those found in northern or southern lower latitudes. The authors provide statistics, but there is no percentage or other parameter that can be quickly appreciated.

We now add an additional multi-panel supplementary figure that shows each protein parameter's relationship to temperature, which is, we believe, more relevant than latitude itself for this type of analysis.

These panels reflect the correlations or lack of correlation for the various parameters shown in the main text and can be quickly appreciated, as requested by the reviewer, and we cite the figure as follows in the main text:

Lines 584–586: Furthermore, examining the average values for these protein parameters according to sample and temperature reflected the patterns observed via other statistical analyses (Supplementary Fig. 8, Supplementary Fig. 9), reinforcing our observations.

We also add the script used for generation of these plots to the github repository.

We add the following Figure legend:

Lines 2050–2053: **S. Fig. 8.** NMDS plots (left) showing the relationships between samples and environmental parameters and Spearman correlations (right) of environmental parameters to amino acid traits for **a**, Caudoviricetes; **b**, Megaviricetes; and **c**, Others (non-Caudoviricetes, non-Megaviricetes).

Lines 2055–2059: **S. Fig. 9.** Correlation of protein parameters with temperature. Each panel shows the relationship of temperature and the mean of the specified protein parameters calculated for each sample from this study for both the Fram Strait (blue) and GOV2.0 (red) datasets. The blue line represents a trend line using LOESS curve fitting with a 95% confidence interval (grey).

L331. The quantity of free viruses cited here were obtained in 2014 during in partial transect compared to the one followed in this work. Although due to the periodicity observed, it is likely that those numbers were similar, the authors could support this by using metaviromes and metagenomes recovered from the same sample and check the vOTUs abundance recovered in both.

We appreciate the reviewer's idea here. The very nice 2014 study focused on amplicon sequencing of g23 and mcp proteins and cell and virus enumeration in free living size fractions. It is a very informative study, but does not allow for easy comparison to our study (or calculating cVCR), because there are no metagenomes (to the best of our knowledge). We searched for updated metagenomes from this study, but did not find any. Nevertheless, we have clarified that our cVCR results do not strictly correlate with their count data, as mentioned elsewhere, and again that our alpha diversity measurements correspond to theirs (which we further clarify is strongly correlated with prokaryotic alpha diversity).

In reference to this paper from 2014, we added new text at Lines 69–71 as indicated above.

L438. RaFAH results are not shown anywhere in the text. The iPHoP results incorporate RaFAH, which helps improve phage-host association predictions. Please, correct the sentence.

This sentence has been corrected as suggested:

Lines 807–808: When multiple host predictions were available, we selected the prediction with the highest confidence score.

L452. Is reference 88 correctly placed here? Please cite the original source where it is described.

In reference 88 the method we utilized is described more thoroughly, and is the first documentation of that method. We now also additionally cite the paper from which the universal single copy genes (in this case, a set of 16 ribosomal proteins) were developed (Hug et al., 2016).

L536. The metagenomes chosen to map the viruses were those obtained using a prokaryotic size fraction of 0.2–3 microns. However, authors do not cite to use any pre-filtering, so, collections with larger sizes could also be suitable. Please, revise the sentence accordingly, as these metagenomes were selected because the authors focused on prokaryotic dsDNA virus relationships.

Lines 964–969: Only DNA samples from discrete depths (e.g., not composites of the mixed layers or similar) and the prokaryotic size-fraction (0.2–3 μm) were used to allow comparison with our dataset. Multiple other size fractions are available, but the 0.2–3 μm was utilized because of our focus on prokaryotic dsDNA viruses, its comparability to the present Fram Strait virus dataset, because it is among the most common range sampled (i.e., many samples available).

Figure 4a. Please, modify this figure to better differentiate viruses from ASVs.

As suggested by Reviewer 2, we have moved this figure to the supplement where it is now approximately 3x as large as previous. We distinguish vOTUs from ASVs better by altering the border width around their symbols. Together, we hope that this helps differentiate vOTUs and ASVs.

Figure 5a. *psbA* gene in italics.

Changed throughout.

Please, enlarge the figures for improved readability. At their current size, it is challenging to distinguish finer details.

We have increased figure size wherever possible to help with distinguishing fine details, and increased the font sizes to improve readability throughout.

Figure S3c. May the authors order the microbial family names accordingly to a taxonomic relationship? It would make more sense than organizing them alphabetically.

We have modified the figure accordingly.

Reviewer #2 (Remarks on code availability):

We thank the reviewer for checking the code. We now upload relevant inputs to our github: https://github.com/alyzzabc/fram_strait_viruses_2016-2020/tree/main/input_tables

Reviewer #3 (Remarks to the Author):

The authors present a well-written paper about a novel dataset that describes a rare 4-year time series of viral metagenomes from the Arctic. They introduce the broad seasonal patterns of viruses they observed and add extra analysis by comparing the viruses they found with other publicly available datasets of that type. They use a fairly straight forward analysis with appropriate and common pipelines and tools.

We appreciate the positive evaluation of our submission and recognition of the value of the dataset.

However, there is one noticeable exception: the authors look at a mix of morphotype classification (podo-, myo- and siphoviridae) for bacteriophages and genetic based taxonomic groups for eukaryote viruses (Phycodnaviridae, Mimiviridae). This seems like an outdated approach since the ICTV reported that they are not monophyletic, phylogenetically solid groups (See e.g. "Taxonomy of prokaryotic viruses: 2018-2019 update from the ICTV Bacterial and Archaeal Viruses Subcommittee"). A direct comparison between those different types of classification groups not ideal, and there is no clear reason why the more recent classification has not been chosen. The main focus of the paper is not on that aspect, so it is not an unfixable problem. But please update the analysis according to the most recent ICTV guidelines, or at least justify why the chosen approach is still valid (I will leave that up to the editor). The authors should at minimum acknowledge this issue in the text and explain why this comparison was chosen, rather than the more up-to-date classification established in recent years.

We have replied to this point at length in response to Reviewer 2, and update the manuscript accordingly throughout. Overall, it required re-generating multiple figures and text values, but has not significantly changed the interpretation of our paper since as most of our main results were framed by the taxonomy as a helpful descriptor, but overall were not a major differentiating factor (i.e., within Caudoviricetes).

It is crucial that the authors carefully go through the entire manuscript and make absolutely sure that they provide a citation for every software tool, package and database and whatever else they use. Studies like this one would not be possible without all the public databases and free software tools developed by other scientists, it is only good practice to give everyone credit for their work.

We agree with the reviewer on this, and apologize for missing some references (highlighted below). In the original version, yes, we had cited many softwares at only the first mention. We now cite also at any repeated mentions, as well as those highlighted below that were missed.

Minor comments:

22-23: Perhaps specify that the paper is about viruses of microbes.

Amended as suggested.

26-28: Wording could be clearer.

We simplify:

Lines 27–28: Among 5,662 vOTUs, 98% and 2% were Caudoviricetes and Megaviricetes, respectively.

62-63: A new paper has been released after the authors would have submitted their manuscript, but there is a viral metagenomic time-series that might be relevant to consider here. <https://academic.oup.com/ismej/advance-article/doi/10.1093/ismejo/wrae216/7833430>

We now cite this paper in the Introduction and Discussion where we compare our results with those from more temperate environments:

Lines 77–81: However, due to the challenges of continuous sampling in the polar regions across multiple years, the degree of seasonality among polar viruses – here, meaning annually repeating patterns of populations and communities across the same

seasons of different years²⁶ – and the potential ecological implications remain to be discovered. Such seasonality has been previously observed in marine temperate environments^{27–29}.

Lines 641–642: However, we observed low similarity in opposing seasons compared to studies from more temperate areas^{27–29}.

78–80: It is known that microbes in the polar regions are highly seasonal and, obviously, virus abundance is directly linked to host abundance, a metric the authors use as well. Given that seasonality of viral communities has been shown in other regions and has been speculated for polar viruses as well, this is not a good hypothesis in my opinion. The authors present a very important and interesting dataset, which as far as I know is the first virome time series from the Arctic Ocean. They should emphasize the novelty of this dataset, rather than the general concept of viral seasonality.

We address this point immediately below, editing the text in accordance with the suggestion.

An additional side point: the manuscript only provides data from the Fram Strait, so the hypothesis should be about Arctic seasonality, not polar seasonality, in any case. It's reasonable to speculate similar patterns for the Southern Ocean, but the seasonal dataset only really covers the Arctic.

We edit this paragraph accordingly.

Lines 118–127: Given this, and the tight coupling of viruses and their hosts^{9–11}, we aimed to examine the degree that viral populations are seasonally structured in the Arctic. The strong seasonal gradients in the Arctic provide an opportunity to observe host-virus dynamics and how they relate to prevailing conceptual models such as the Piggyback-the-winner^{42–44}, Constant-Diversity^{45,46}, and Red-Queen^{28,47} hypotheses. We examined the diversity and seasonality of dsDNA virus communities in the Arctic Fram Strait over four complete annual cycles at roughly monthly resolution. This unprecedented time-series of virus communities in the Arctic demonstrates considerable seasonality in viral diversity, their association with environmental conditions and potential microbial hosts, and their distribution across the global ocean.

90–91: What was the depths these samples were collected from? Please clarify.

Lines 131–133: We examined 47 long-read metagenomes from samples collected at near-monthly resolution over a four-year period (Aug 2016 – Aug 2020) in the WSC (Fig. 1a, Supplementary Data 1), from an average depth of 29 m.

Lines 737–739: Moored Remote Access Samplers (RAS; McLane) autonomously collected and fixed seawater from an average depth of 29 m in the eastern and western Fram Strait (Fig. 1) at weekly to fortnightly intervals between 2016–2020^{41,58}.

117: s missing for ratios?

Fixed.

166: Please introduce the meaning of abbreviation PAR.

The abbreviation PAR has been defined as suggested (Lines 273–274).

162-171: Please provide a value or a figure for this. The stats are buried in a difficult-to-read supplementary table, but this is a fairly important point and the following paragraph is building on that point. It should be made clear to the reader how significant that correlation is compared to oxygen and MLD. The authors provide a main figure (3) and a percentage to show the results for the less important physiochemical drivers, but not for the correlation with their hosts. The text and figures should reflect the more important aspects of results first.

We have included the Mantel ρ statistics in the text Lines 268-271, and added Supplementary Fig. 4 which shows the complete Spearman correlations together with the Mantel test.

We cite this figure in the main text as follows:

Lines 268–271: At the whole community level, Mantel tests demonstrated the strongest correlation to prokaryotic community composition (Mantel $\rho = 0.632$, $p = 0.001$) followed by oxygen (Mantel $\rho = 0.371$, $p = 0.001$) and mixed layer depth (MLD) (Mantel $\rho = 0.259$, $p = 0.001$) (Supplementary Data 3, Supplementary Fig. 4).

241-242: This is not surprising since the authors group the viruses based on the outdated morphological classification, which is polyphyletic.

To improve clarity (as mentioned elsewhere) we have removed this figure panel and the associated text, and replaced it with a simpler figure.

287: Define what depth is considered deep ocean. Based on context, > 200m depth.

We clarify:

Lines 537–538: (here, operationally, sampling depth of greater than than 200 m)

467: Please cite the authors of the vegan and ggplot2 packages and acknowledge the use of R.

The citations have been added as requested for the use of R (Lines 917–918), vegan (Line 862, Line 918, Line 1026, Line 1029), and ggplot2 (Line 864, Line 1033).

501: Please reference the original Cytoscape paper.

For the original Cytoscape paper, Line 911.

518: Please cite the original vConTACT2 paper.

For the original vConTACT2 paper, Line 927.

519: Please cite the hmmer paper.

For hmmer, Line 928.

520: Please add citation for UniRef.

For UniRef, Line 930.

532: Please add citations for Tara Oceans, Malaspina Expedition and Bio-GO-SHIP.

They are hidden in a supplementary table as a link only, but should be properly cited in the text itself too. I will stop checking if the authors have referenced every tool, database and package, since they tend to only add a citation upon first mention, if at all, so it's tedious and easy to miss.

We now cite these main large-scale studies in the main text, and the rest we used data from are now referenced in the Methods:

Lines 499–502: ...we assessed the distribution of Fram Strait viruses across the global oceans through metagenomic datasets from various large-scale sampling campaigns, such as Malaspina, Tara Oceans and Bio-GO-SHIP^{14,65–69} (Supplementary Data 7).

Lines 961–962: Metagenomes from ten oceanic datasets^{14,65–69,139–144}, including Tara Oceans, Malaspina, and Bio-GO-SHIP (Supplementary Data 7), were downloaded from NCBI,....

Reviewer #3 (Remarks on code availability):

The codes seem fine, there is enough instructions and explanations.

Thank you for the positive assessment of our code availability.

Additional edits

We also updated a few methodological aspects related to the network analysis: We adjusted the Git Repository link (<https://gitlab.com/qtb-hhu/marine/publications/framphages2024>), cited an implemented GUI-based framework for community dynamics using network analysis called Otter (DOI 10.5281/zenodo.13840702 (<https://zenodo.org/records/13840807>), and a recent paper describing the method (Beyond blooms: the winter ecosystem reset determines microeukaryotic community dynamics in the Fram Strait. <https://www.nature.com/articles/s43247-024-01782-0>)

REVIEWER COMMENTS

Reviewer #1 (Remarks to the Author):

I'd like to thank the authors for their efforts in addressing my comments and improving the manuscript. After reading the rebuttal letter and the revised manuscript, I find that most of the responses are adequate, and the revisions are well-explained and feasible. The improvements, such as clarifying genome completeness and expanding discussions on arctic adaptations at gene level, are appreciated. Overall, the manuscript has largely improved.

Thank you for the help in improving our manuscript.

I still have a few remaining suggestions to enhance robustness, transparency, and clarity of this paper.

1. First of all, I have to say sorry that I made a mistake in my first revision about the total sequencing depth. I originally thought it was 10 Gb per sample, but, in fact, it is only 1 Gb (196,489 reads × 5,435 bp per read). This low sequencing depth makes the quantification of viruses more challenging than I expected, especially given that viral diversity does not reach saturation in the dataset.

The updated results indicate that viral diversity peaks in late summer to autumn, which is the somehow opposite of what was stated in the initial submission. This is a big shift in the overall viral seasonal dynamics.

My question is:

The error bars in Figure 2B are very large, making it difficult to compare diversity across months with confidence. Would it be possible to restrict the analysis to only samples with adequate viral reads when calculating diversity? Alternatively, it might be helpful to present a fully expanded diversity/richness plot for each year, similar to what was done for cVCR in Fig.1 and Fig.S1.

We appreciate the reviewers point here, we have now replaced the barchart with error bars with a fully expanded diversity/richness plot as was done for cVCR in Fig. S1, as requested. This shows that there is intra- and inter-annual variability in richness, which was contributing to the large error bars, and this version indeed provides for better context and interpretation.

Notably – and this applies to all points related to the alpha diversity assessments – immediately below, as well as Point #4. We realized that when we computed the total

viral read numbers we did so incorrectly (just a downstream coding misinterpretation of a mapping table). After re-calculating this value, which does not affect the cVCR or cVGB (calculations which were done independently of this explicitly calculated value), the total mapped reads are actually $8,825 \pm 1,020$ virus reads per sample. The main impact on the alpha diversity aspect was it allowed us to set higher sub-sampling (see below) which provided an overall more robust assessment of the alpha diversity, though of course still limited by relatively low-throughput Pac Bio relative to Illumina libraries. We discuss more where relevant, below.

In addition to the alpha diversity assessments, it also impacted Supplementary Table (Tiara results), and a line in the text. See Point #4 below for more information about this.

L119-123: On average, each sample harbored $9,815 \pm 965$ viral reads based on raw read counts ($5.3 \pm 0.5\%$), and $8,825 \pm 1,020$ reads mapping to vOTUs ($6.8\% \pm 1.8\%$). Notably, despite not pre-filtering to remove eukaryotes, the large majority of reads in the dataset were prokaryotic ($90.2\% \pm 1.4\%$), with eukaryotic reads making up a smaller proportion ($7.2\% \pm 1.0\%$).

Also, I have a question regarding the subsampling approach. As described, subsampling was performed with “100 iterations of subsampling the vOTU count table at a range of different depths, spanning from 25 up to 16,000 counts at 50-count intervals.”

Does the 25% mapping breadth threshold still apply in this subsampling process? Specifically, if removing just one read causes the mapping coverage to drop below 25%, would that vOTU still be considered? The method here is important, particularly to richness estimation, where rare viruses may be affected. I suggest that the authors provide more details on how subsampling was performed with the mapping threshold.

We appreciate the reviewer’s question. Only contigs that had 25% breadth of coverage considering all the data available for a given sample were considered for the analysis. From that point, we did the sub-sampling (which was able to go to a higher value since we corrected the total sequence numbers, as described above).

We now provide more details about how subsampling was performed in the methods.

L762-776: For this, we applied 100 iterations of subsampling the vOTU count table at a range of different depths, spanning from 50 up to 30,000 counts at 50 count intervals. For this step, all vOTUs that exceeded the 25% breadth of coverage cutoff when using all the data for a given sample were utilized. During each iteration, richness, evenness and Shannon diversity was calculated, with mean values being determined for each 50-

count interval. The functions *rrarefy*, *specnumber* and *diversity* from the *vegan* package¹²² were used to perform subsampling and alpha diversity calculations. The mean values were used to generate a rarefaction curve of alpha diversity using the *ggplot2* package¹²³. To assess shifts in diversity over time, we compared the mean richness values obtained from subsampling at the 1000 count interval. In addition, we computed and compared the slopes of the sample rarefaction curves up until the subsampled depths, which represents the rate of vOTU discovery with increasing viral read counts. Comparing the slopes across samples enabled an assessment of whether the sample richness patterns observed at the subsampled depth would change if the viral read count was increased.

In addition, the usage of slope and direct subsampling points to compare diversity/richness across sequencing depths might be improved by using non-linear functions. For example, combining rarefaction and extrapolation using the iNEXT R-package could provide estimates of diversity/richness accounting for undetected viruses beyond observed data.

We appreciate the reviewers point here as well. Given the coverage of the long-reads, and that it is not yet approaching saturation, we had preferred not to extrapolate the curves. Also for this reason, we think that showing the accumulation curves is valuable. However, we did recalculate the linear modelled values, after reviewing the reviewers point we realized it would be better to calculate it only until 1000 reads (the subsampled depth used in Fig 2c. In addition to this, we applied **iNEXT which shows results very similar to the trends we observed from the linear approach**. Both approaches show lowest diversity estimates in the spring, and higher diversity in the summer. We clarify this mainly in Fig 2. Legend, along with adding the iNEXT results as Supplementary Fig 2. Also we update the methods to state how the iNEXT was employed.

L1584-1594: Intra-and inter-annual viral richness variability. a, To assess differences in sequencing depth, we iteratively subsampled viral read counts from 50 up to 30,000 at 50 count intervals and at each interval determined the mean richness from 100 iterations. The mean richness across subsampled intervals was visualized in a rarefaction-style curve. b, Boxplots illustrating the slope of the vOTU richness rarefaction curves up until the chosen subsampling depth (1000 viral reads). The slopes represent the trajectory of vOTU discovery, and thus indicate how richness may appear if a higher sequencing depth was achieved. c, vOTU richness at subsampled depth across sampling months. Viral read counts were subsampled to 1000 per sample and vOTU richness calculating through 100 iterations. vOTU richness and standard deviation from the 100 iterations are illustrated through the bar and error bars, respectively.

iNext Methods, L776-778: To further explore this, we also employed the iNEXT v3.0.1 package^{125,126} to estimate the vOTU richness based on extrapolations to 30,000 viral read counts.

Given the big change in diversity results, I wonder whether another key metric, cVCR, and its associated results are also influenced by sequencing depth and normalization (subsampling). Since cVCR relies on the coverage of both viral and cellular genomes, low and varied sequencing depth could potentially bias cVCR values as well. So subsampling may potentially introduce some shifts in results, particularly in seasonal trends presented in Fig 1.

I believe the observed cVCR-based abundance seasonal pattern is biologically meaningful, but I would prefer authors to reinforce the conclusions by addressing these potential biases.

My suggestion is to justify the use of cVCR by testing whether cVCR values remain stable when reads are subsampled to the same sequencing depth.

We appreciate the reviewer's concern here, and to address it, we randomly sub-sampled all samples to 100,000 reads and re-performed the cVCR analysis. This excluded seven samples with fewer than 100,000 total reads. The results are consistent with those from the non-sub-sampled dataset in Fig. 1 and Fig. S1. There are some month-to-month differences in the cVCR vs the full dataset, which would be expected to stochastic variation and sometimes low variation in viruses in these sub-samples. **Encouragingly, the overall magnitude of cVCR and the overall seasonal trends with this sub-sampling approach are highly similar to when using the entire dataset.** We now provide a new panel to Fig. S1 which shows this results, and cite it in the results, adding also a detail about an alternative method we calculated, coverage per Gigabase pair.

L158-161: These patterns were consistent also when we sub-sampled the total reads to 100,000 reads per sample, as a test of the robustness of this metric to sequencing depth, as well as when we calculated coverage per Gigabase (Supplementary Fig. 1); given these consistencies, we focus the remaining analyses on cVCR from the full dataset.

2. As far as I understand, the parameter "-B 1000" used in the phylogenetic analysis, as written in the Methods section, refers to UFBoot. This should be explicitly stated in the Fig.5 legend, because different from traditional values, UFBoot values are only

considered reliable if they are $\geq 95\%$ (<http://www.iqtree.org/doc/Frequently-Asked-Questions>).

We have now labeled the branches in accordance with the reference that the reviewer suggested (SH-aLRT $> 95\%$ and Ultrafast Bootstraps > 95), as shown in the figure.

Additionally, it would be greatly appreciated if the authors could provide the full psb phylogenetic tree and alignment as supplementary files to allow readers to evaluate evolutionary relationships.

Also we have uploaded the full sequences, alignment, trimmed alignments, and treefile to figshare and add it to the data availability statement: [10.6084/m9.figshare.28398173](https://doi.org/10.6084/m9.figshare.28398173)

L1030-1032: psbA alignment, trimmed alignment, treefile, and labels for cyanophage references are available via figshare (DOI [10.6084/m9.figshare.28398173](https://doi.org/10.6084/m9.figshare.28398173)).

We also corrected the heatmap legend -- in editing the figure for the previous re-submission we had mistakenly made the values 100x too high. It is now consistent with the original submission, and the text, which was correct in each version.

#3. The paper [<https://pubmed.ncbi.nlm.nih.gov/39448562/>] provides a background on polar viruses in a time-series context. The authors may consider referencing it to strengthen the introduction and discussion.

We add to the Introduction: L58-60: In addition to spatial structuring, polar viral communities also shift over time, for example in the Antarctic, following bloom dynamics and assembling into distinct communities across seasons²³.

We add to the Discussion: L513-515: In terms of total virus prevalence, we found strong increases in the ratio of viruses to prokaryotes in the late summer months, mirroring findings from the Antarctic where viruses peaked in mid to late summer²³.

#4. I need some clarification for the newly added values in LN 145 because “On average, each sample harbored $10,215 \pm 1,116$ viral reads” (LN 142) and “The total number of reads per sample was $196,489 \pm 18,358$ ” (LN 138).

If the total reads per sample are $\sim 196,489$, how is the virus read proportion only $0.4\% \pm 0.5\%$? Additionally, when I checked Supplementary Table 1, the average number of viral reads per sample is ~ 800 (Column AN), which does not match the reported $10,215 \pm 1,116$.

Could the authors clarify whether there is an inconsistency in these values?

As noted above for point #1, we appreciate the reviewers point here which made us realize that indeed there was an inconsistency in the values between the text and Supplementary Data 1. The virus read values in Supplementary Data 1 were incorrect (just a misinterpretation based on a downstream column of mapping data). We have recalculated the aspects of the paper in which this was relevant, namely the line in the text referenced by the reviewer, Supplementary Data 1, and the Tiara results (in which we remove viral reads before calculating, since Tiara has no explicit viral prediction), and viral richness maximum rarefaction values. Fortunately, no other aspects were impacted by this inconsistency.

L119-123: On average, each sample harbored $9,815 \pm 965$ viral reads based on raw read counts ($5.3 \pm 0.5\%$), and $8,825 \pm 1,020$ reads mapping to vOTUs ($6.8\% \pm 1.8\%$). Notably, despite not pre-filtering to remove eukaryotes, the large majority of reads in the dataset were prokaryotic ($90.2\% \pm 1.4\%$), with eukaryotic reads making up a smaller proportion ($7.2\% \pm 1.0\%$).

#5. Fig. 2 needs clearer descriptions. For example, what does the error bar mean in the Fig.2B? And the of Fig2B seems to be “mean of richness values in subsampling” as I understand, but the legends said “b, Slope of richness rarefaction curves across sampling months.”, which is overlapped with the Fig S2.

As previously mentioned above, we have revised the Figure 2 legend for clarity and added another panel to Fig. 2. The updated Fig. 2 now includes (a) observed richness of vOTUs shown as rarefaction curves, (b) summary of the slopes of the rarefaction curves as estimates of overall richness and (c) vOTU richness per sampling time point presented as bar plots at a depth of 1000 reads per sample. We believe that the revised figure is more clear and robust over the previous versions, and thank the reviewer for their help in improving it.

Full Fig. 2 legend (Line 1584-1594) is listed above as part of point #1

Other legends could also be improved by providing more details. For example in Fig8A, what input data was used to calculate NMDS?

We now update Fig. 8A stating what input data was used to calculate NMDS, as well as providing some clarification in the results section related to 8a and other related figures, which we believe clarify the interpretation.

L1650-1662: Fig. 8. Viral amino acid traits across environmental gradients. a, Nonmetric multidimensional scaling (NMDS) plot showing the environmental diversity of the sampled Fram Strait and GOV2.0 ecosystems. The Bray-Curtis distance similarity matrix was calculated based on the available environmental parameters of 38 Fram Strait and 69 GOV2.0 samples and used to generate NMDS coordinates of each sample. Point shapes represent dataset origin, and colors distinguish different temperature ranges. Vectors show correlations with environmental variables. This ordination provides context for the correlations to environmental parameters of amino acid traits shown in part b. b, Heatmap plot illustrating the Spearman correlation coefficients of environmental parameters to amino acid traits, with p-values represented by asterisks as indicated. The Spearman coefficients were calculated using a Mantel test using pairwise distances of each environmental parameter (Euclidean distance) vs. each amino acid trait (Bray-Curtis distance).

We also amended the results section to clarify that the NMDS is based only on environmental data which is meant to provide the environmental overview/context and a sense of the diversity of the samples surveyed, in the correlations shown in the Figure 8b. Also, we more directly cited the complementary Figures in this section, which breakdown these correlations by family in the supplement.

L460-476: To ensure fair comparisons between the datasets, we focused only on GOV samples originating from ≤ 35 m water depth (i.e. the average depth of Fram Strait sampling) and first analysed the environmental context, which shows strong clustering of GOV2.0 polar samples to the Fram Strait, based on their similarity in distance to equator, oxygen, and temperature (Fig. 8a). Linking those environmental data to the examined protein features for all viruses, revealed that the aliphatic index and the nitrogen usage score were positively correlated with temperature (Fig. 8b), whereas other traits of potential cold adaptation, in particular polar charged and uncharged amino acids, showed significant correlation with one or more of the other environmental parameters, but not directly with temperature. This was also the case for *Caudoviricetes*, while *Megaviricetes* amino acid traits were more generally positively correlated with oxygen, temperature, and chlorophyll, and 'other' vOTUs were generally positively correlated with salinity and oxygen (Supplementary Fig. 9). In the latter cases, these groups are more poorly sampled due to their low abundance resulting in weaker correlations overall, and especially necessitating further study. Examining the average values for these protein parameters according to sample and temperature, across all viruses, reflected the patterns observed via other statistical analyses (Supplementary Fig. 10), reinforcing our observations.

6. Viral taxa should be written in italics throughout the manuscript.

All mentions of viral taxa have been italicized.

We have now also italicized prokaryotic Families, Orders, etc (in accordance to recent Nat. Comm. papers and Parker CT, Tindall BJ, Garrity GM. International code of nomenclature of prokaryotes. Prokaryotic code (2008 Revision) International Journal of Systematic and Evolutionary Microbiology. 2019;69(1A):S1–S111. doi: 10.1099/ijsem.0.000778.

Reviewer #2 (Remarks to the Author):

Review Nature Communications. NCOMMS-24-55589

Arctic Ocean virus communities: seasonality, bipolarity, and prokaryotic associations

I would like to sincerely thank the authors for their thorough and thoughtful responses to not only my comments but also those of the other reviewers. Their efforts to address the feedback provided have significantly improved the quality of the manuscript, and I appreciate the time and attention they dedicated to refining their work.

Thank you for your help in improving our manuscript.

However, I would just like to highlight the following points:

L 488-490. Please clarify the following statement: “Notably, in the entire dataset of ASV to vOTU correlations, only one had a predicted host of Nitrosopumilaceae, which was correlated with an unclassified Gammaproteobacterium (no family-level prediction)”. Does this mean that one virus has two putative host assignments, one being an archaeon (Nitrosopumilaceae) and the other a Gammaproteobacterium?

We have tried to clarify this:

L390-395: Notably, among vOTUs that had correlations to hosts in the correlation network, only one had a predicted host (based on iPHoP) of *Nitrosopumilaceae*. This vOTU was correlated with an unclassified Gammaproteobacterium (no family-level prediction) within the correlation network, highlighting again the difficulty comparing directly these two complex and independent approaches.

The discussion about the finding that many of Fram Strait viruses are also abundant in deep samples could include also the possibility that ocean currents in specific areas play an important role. The reason why these viruses are found at such depths remains puzzling.

We changed one line of the results to be more precise, i.e., from 'detected commonly' vs 'common' which may imply more strongly that they are abundant, vs just observed.

We also add to the Discussion:

L584-598: Additionally, in general, the detection of the Fram Strait viruses in the deep ocean is notable, and is potentially related to export from the surface and deep ocean currents, some of which originate at the Arctic. Further inspection into the specific vOTUs and from other datasets at the global level could help further evaluate these ideas. Part of the explanation may be that it is vOTUs we primarily observed in the deep waters are from M6, the 'persistently present' Fram Strait module, thus, they may be associated also with generally persistent/ubiquitous host lineages and/or have broad host ranges, and furthermore the relative lack of seasonality in the deep sea may help with their consistent detection. In the discussion, we add:

L581-598: Further investigations to unravel the diversity and dynamics of viruses in Arctic and Antarctic are crucial, given that climate shifts are causing major biological and physicochemical perturbations in these regions. Additionally, in general, the detection of the Fram Strait viruses in the deep ocean is puzzling, and is potentially related to export from the surface and deep ocean currents, some of which originate at the Arctic. Further inspection into the specific vOTUs and from other datasets at the global level could help further evaluate these ideas. Part of the explanation may be that it is vOTUs we primarily observed in the deep waters are from M6, the 'persistently present' Fram Strait module, thus, they may be associated also with generally persistent/ubiquitous host lineages and/or have broad host ranges, and furthermore the relative lack of seasonality in the deep sea may help with their consistent detection.

Additionally, the term 'epipelagic' may be more appropriate than 'surface waters' when referring to depths of less than 200 m.

We have changed surface waters to epipelagic throughout the manuscript.

Regarding the new discussion paragraphs on how well these findings fit into host-virus evolutionary models, I completely agree with the authors that ecological models are

often idealized in a chemostat-like setting. These dynamics primarily occur within seasons and should be evaluated through more spatially resolved temporal sampling.

However, I would like the authors to conduct a simple analysis: checking whether the exact same virus appears in consecutive summers, springs, or winters (not at 95%, if not at 100% ID). Are there any viral sequences with no genetic changes or SNPs observed across the temporal samples? The “Red Queen model” hypothesizes that ‘fast’ co-evolution occurs between hosts and viruses. If no genetic changes are detected, this hypothesis could be ruled out, at least within the analyzed period and possibly for a particular type of viruses. Instead, this would suggest the presence of a “viral seed bank” that persists in the environment between seasons but remains undetected due to the low abundance of its host at the change of season, influencing biodiversity in a way perhaps more aligned with the constant-diversity model. However, further analysis would be needed to confirm this hypothesis.

We thank the reviewer’s thoughtful consideration of the discussion we wrote in response to their previous suggestions, and are glad that they appreciate it.

In regards to the new analyses proposed, while a comprehensive analysis in this direction would be somewhat outside the scope of our already quite long paper, we have considered and addressed it in two ways. First, we explicitly compared viral contigs >10,000 bp across years. Out of 7,360, we found 21 clusters of contigs, encompassing 55 contigs total, that were 100% matches (akin to 100% vOTUs) between at least two years. This suggests that detection of viruses with no genetic changes across years is rare, though of course there is a limitation of sequencing depth. Second, we employed a state-of-the-art tool, inStrain (<https://www.nature.com/articles/s41587-020-00797-0>), which compares reads mapped to vOTUs, and examines for SNPs. We used 98% minimum ANI, the most stringent threshold that the developers recommend, and a minimum coverage of 5 (default setting) to call a variant. Then we filtered for contigs where more than 25% of their bases were covered by at least one read, i.e. $\text{breadth} > 0.25$. To confirm a SNV, we required a minimum SNV frequency of 0.05 and a false discovery rate of $1e-06$ (default settings). Among the 202 comparisons (i.e., enabled by coverage exceeding minima required to detect SNPs), we find that there tends to be an underlying diversity within the vOTU populations as reflected by inStrain’s consensus SNP metric, though the amount varies between values that are likely insignificant (in accordance with the documentation (>0.9999) to about 0.97. This highlights that, in general, vOTUs are made up of populations of highly similar viruses, rather than being clonal. This has also been observed elsewhere, including the arctic.

Therefore, we cannot rule out the “Red Queen Model”, and we can also not rule out the constant-diversity model without more detailed and comprehensive temporal and spatial data.

As this analysis is outside the original scope of our paper and is thus not a comprehensive interrogation of the strain diversity which would indeed be a valuable future direction, we add two sentences in the alpha diversity section of the manuscript related to the new analysis which also enable us to add a point to the discussion. Also we add the inStrain results as a supplementary figure.

Results, L211-222: To further contextualize the dynamics we observed at the vOTU level, we examined the underlying sequence diversity based on examining exact contigs across years and single nucleotide variation. First, we explicitly compared viral contigs >10,000 bp across years. Out of 7,360 contigs, we found 21 clusters of contigs, encompassing 55 contigs total, that were 100% matches between at least two years. This suggests that detection of viruses with no genetic changes across years is rare, though there is a limitation of sequencing depth. Second, based on analysis of SNPs in reads to vOTUs, we find that there tends to be an underlying diversity within the vOTU populations, though the amount varies between vOTUs ranging from likely insignificant variation related to sequencing error, to high differentiation (e.g., 97% similarity) (Supplementary Fig. #). These analyses highlight that, in general, vOTUs are made up of populations of highly similar viruses, rather than being clonal. This has also been observed elsewhere, including the Arctic¹⁴.

Discussion, L623-629: Our analysis of microdiversity of the Arctic virus communities demonstrated both instances of identical viruses across years, as well as a general tendency for vOTU to be underlaid by sequences with very high similarity but not exact matches to the vOTUs, akin to that seen elsewhere in the ocean^{29,30,88}, including the Arctic Ocean¹⁴. Thus, based on these results we cannot rule out either the Red-Queen hypothesis or Constant-diversity.

I have no other questions for this manuscript.

Reviewer #2 (Remarks on code availability):

The code is well-organized and easy to follow, which makes it a valuable resource for the community.

Thank you.

Reviewer #3 (Remarks to the Author):

The authors have addressed all comments to a satisfactory degree. I have no further concerns or suggestions and recommend this manuscript for publication.

Thank you.